



# Contrail formation for aircraft with hydrogen combustion - Part 1: A systematic microphysical investigation

Josef Zink[1], Simon Unterstrasser[1], and Ulrike Burkhardt[1]

[1]Deutsches Zentrum für Luft- und Raumfahrt, Institut für Physik der Atmosphäre, Oberpfaffenhofen, Germany

**Correspondence:** Josef Zink (josef.zink@dlr.de)

**Abstract.** The number of ice crystals formed during the contrail's jet phase has a long-lasting impact on the life cycle and radiative forcing of contrail cirrus clouds. Contrail formation for conventional kerosene combustion is well studied, and suitable parametrizations for the early ice crystal number have been used to estimate the climate impact of contrail cirrus with a general circulation model. However, a parametrization for the number of ice crystals formed is lacking for hydrogen combustion. To develop such a parametrization, we present a comprehensive set of contrail formation simulations using the particle-based Lagrangian Cloud Module in a box model approach. Unlike kerosene combustion, no soot particles are emitted in the hydrogen combustion case. Thus, ice crystal formation is assumed to occur on ambient aerosols entrained into the exhaust plume. The results show that coarse mode particles have negligible influence on ice crystal number due to their low abundance. Furthermore, ice crystal formation involving multiple co-existing aerosol populations with different properties (number, size, solubility) can be reconstructed from simulations involving single aerosol populations. We also identify atmospheric conditions where homogeneous droplet nucleation can be safely neglected as potential ice formation pathway. Based on more than 20,000 simulations covering a broad range of atmospheric conditions and aerosol properties, we identify a regime where ice crystal formation becomes nearly independent of ambient relative humidity, aerosol size, and solubility. Our results provide a basis for a data-driven parametrization of ice crystal number in contrails from hydrogen combustion, to be presented in a companion paper.

## 1 Introduction

### 1.1 Motivation

A hydrogen economy envisions the widespread use of hydrogen as a clean energy carrier, replacing fossil fuels in sectors such as transportation, industry, and power generation (Staffell et al., 2019). The successful implementation of a hydrogen economy requires large-scale infrastructure for hydrogen production, storage, distribution, and end-use technologies such as fuel cells or gas turbines (Tashie-Lewis and Nnabuife, 2021). Hydrogen produced via electrolysis using renewable energy (commonly referred to as green hydrogen) is free of carbon emissions during its production (Dincer and Acar, 2015; Nikolaidis and Poullikkas, 2017; Ajanovic et al., 2022). Furthermore, hydrogen used in gas turbines or fuel-cell systems produces no direct greenhouse gas emissions except water vapor.





Despite its potential to have net-zero greenhouse gas emissions, the use of hydrogen is not without environmental concerns. One of the risks involves hydrogen leakage, which can indirectly contribute to global warming by extending the atmospheric lifetime of methane and increasing tropospheric ozone and stratospheric water vapor levels (Ocko and Hamburg, 2022; Hauglustaine et al., 2022; Warwick et al., 2023).

The use of hydrogen in aviation comes with several challenges. On the ground, it requires the development of dedicated production, liquefaction, and storage infrastructure at airports, which demands significant investment and logistical coordination (Hoelzen et al., 2022). Onboard the aircraft, hydrogen's low volumetric energy density necessitates larger and heavily modified fuel tanks, reducing payload capacity or range unless aircraft are redesigned (Tiwari et al., 2024; Soleymani et al., 2024). Cryogenic storage of liquid hydrogen to minimize its volume adds another technical complexity. Despite these challenges, the aviation industry views hydrogen as having great potential for decarbonizing aviation in the long term (Airbus, 2020).

However, the climate impact of aviation extends beyond $CO_2$ emissions, with non-$CO_2$ effects such as $NO_x$ emissions and contrails contributing to the overall radiative forcing. According to Lee et al. (2021), these non-$CO_2$ effects currently account for over half of aviation's total effective radiative forcing, although this estimate is associated with a large uncertainty. Therefore, to properly assess a potential new aviation technology such as hydrogen combustion from the perspective of non-$CO_2$ effects, a solid understanding of all relevant processes is required. This includes an understanding of processes ranging from the contrail formation to the impacts of long-lived contrails.

## 1.2 Contrail life cycle and radiative impact

The life-cycle of a single contrail is typically divided into three regimes (Paoli and Shariff, 2016): During the *jet regime*, hot exhaust air rapidly mixes with the cold ambient air. This mixing is driven by the strong shear layer between the jet and the environment and is characterized by strong turbulence. During this phase, ice crystals may form, typically completed within a few seconds (e.g., Kärcher and Yu, 2009; Lewellen, 2020; Bier et al., 2022, 2024; Yu et al., 2024). The ice crystals formed are then trapped in the cores of the counter-rotating pair of wing-tip vortices induced by the lift generation. During the *vortex regime*, the vortices descend due to their mutual interaction (Spalart, 1998; Gerz et al., 2002). Adiabatic heating resulting from this downward motion causes partial sublimation of the ice crystals (Sussmann and Gierens, 1999; Unterstrasser, 2008; Kleine et al., 2018). Additionally, detrainment from the descending primary wake driven by baroclinic vortex instabilities, gives rise to a secondary wake near the emission level (Lewellen and Lewellen, 1996; Unterstrasser et al., 2014). After vortex-breakup at around $5\,\mathrm{min}$ (sometimes also referred to as an additional phase, the *dissipation regime*), the *dispersion regime* follows associated with the evolution into contrail cirrus (Schumann, 2012; Lewellen, 2014; Unterstrasser et al., 2017a). This regime is marked by contrail spreading due to vertical wind shear, ice crystal growth/sublimation, and ice crystal sedimentation. For suitable ambient conditions, this regime can last several hours until dissolution of the contrail cirrus cloud (Jensen et al., 1998; Haywood et al., 2009; Laken et al., 2012; Bier et al., 2017).

The direct contributions to the contrail's total lifetime-integrated radiative effect during the jet and vortex regimes are typically small (Unterstrasser and Gierens, 2010a). However, early processes occurring in these phases have a substantial impact on the properties and life cycle of the contrail cirrus cloud (Burkhardt et al., 2018). Consequently, these initial processes exert





a significant indirect influence on the overall radiative effect associated with a contrail (Bier and Burkhardt, 2022). While the total ice mass after the vortex break-up has a low impact, the number of ice crystals at this stage strongly and non-linearly in-
fluences the radiative effect of the contrail cirrus cloud for a given meteorological condition (Unterstrasser and Gierens, 2010b; Lottermoser and Unterstrasser, 2025).

The idealized study of a single contrail's life cycle provides helpful insights into the relevant physical processes that influence its radiative impact. However, it cannot deliver an estimate of the radiative impact of the globally distributed contrail coverage, which varies across time and space. This requires a realistic representation of aircraft traffic and the interaction of
the contrails with the background atmosphere. For such purposes, general circulation models (GCM) have been extended to include a parameterization for contrail cirrus (Burkhardt and Kärcher, 2009; Gettelman and Chen, 2013; Bock and Burkhardt, 2016; Schumann et al., 2015). However, due to the coarse spatial and temporal resolution in a GCM, early processes during the jet and vortex phases can not be resolved. Therefore, contrails are typically initialized after the vortex break-up, while earlier processes need to be parametrized. For conventional kerosene combustion, a parametrization for contrail ice nucleation
(Kärcher et al., 2015), valid for the soot-rich regime, and a parametrization for the ice crystal loss during the wake vortex phase (Unterstrasser, 2016) have been implemented in the contrail initialization within ECHAM-CCMod (Bier and Burkhardt, 2019, 2022). Moreover, the parametrization for the ice crystal loss has been recently extended to also include hydrogen combustion scenarios (Lottermoser and Unterstrasser, 2025) ready to be implemented. To date, a suitable parameterization for the number of ice crystals formed during the jet phase does not exist for hydrogen-powered aircraft.

**1.3  Contrail formation for the hydrogen combustion case**

During the jet phase, the mixing of the hot and humid exhaust (called plume) with cold ambient air can lead to transient water supersaturation for suitable ambient conditions. In the absence of micro-physical processes, the plume partial water vapor pressure and temperature evolve along the so-called mixing line (Fig. 1), whose slope can be determined by

$$G = \frac{EI_\mathrm{v} \, c_\mathrm{p} \, p_\mathrm{a}}{\epsilon \, Q (1-\eta)} \; . \tag{1}$$

In Eq. (1), $EI_\mathrm{v}$ is the mass-specific water vapor emission index, $Q$ the combustion heat (lower calorific value), $c_\mathrm{p}$ is the specific heat capacity at constant pressure, $p_\mathrm{a}$ the ambient pressure, $\epsilon = 0.622$ the ratio of molar masses of dry air and water vapor and $\eta$ the overall propulsion efficiency (Schumann, 1996). The slope of the mixing line influences the Schmidt-Appleman threshold temperature (Schmidt, 1941; Appleman, 1953; Schumann, 1996). The Schmidt-Appleman threshold temperature $\Theta_G$ is defined as the ambient temperature for which the mixing line just touches the saturation curve over water for a given ambient relative
humidity over water $RH_\mathrm{wat,a}$ (Fig. 1). If the ambient temperature is below $\Theta_G$, transient water supersaturation is generated during the mixing process and plume particles may activate into liquid droplets.

Burning hydrogen instead of kerosene results in a higher energy-specific water vapor emission index $EI_\mathrm{v}/Q$ of about 2.6 (Schumann, 1996; Bier et al., 2024), thus leading to a steeper slope of the mixing line (Fig. 1). This steeper slope leads to higher and longer-lasting supersaturations compared to conventional kerosene combustion, influencing the contrail formation
process (Bier et al., 2024). In particular, the formation of ice crystals is limited by the homogeneous freezing temperature of



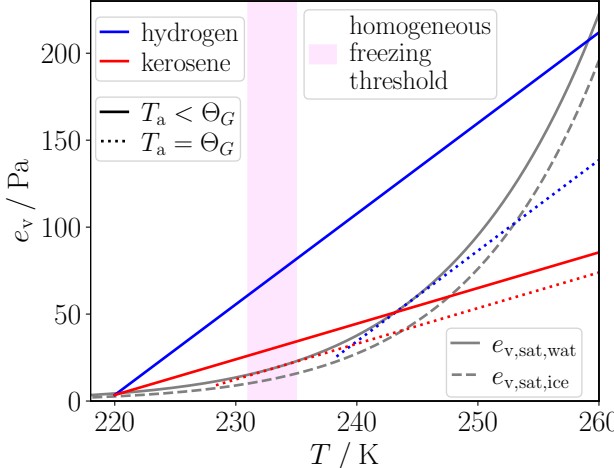

**Figure 1.** Mixing lines (partial water vapor pressure $e_v$ vs plume temperature $T$) are depicted for both hydrogen and kerosene combustion cases for the ambient pressure $p_a = 260\,\mathrm{hPa}$, ambient relative humidity over water $RH_{\mathrm{wat,a}} = 80\,\%$ and overall propulsion efficiency $\eta = 0.4$. Shown are threshold cases with ambient temperatures $T_a = \Theta_G$ such that the mixing line just touches the saturation curve over water $e_{v,\mathrm{sat,wat}}$ for the given $RH_{\mathrm{wat,a}}$ value. Additionally, cases with a lower ambient temperature of $T_a = 220\,\mathrm{K}$ are displayed. Moreover, the range of homogeneous freezing temperatures for supercooled droplets is indicated.

supercooled droplets, which is typically lower than the classical Schmidt-Appleman threshold temperature in the hydrogen combustion case (Fig. 1). Nevertheless, the extent to which the ambient temperature is below the Schmidt-Appleman threshold temperature still plays an important role, as it determines the maximum supersaturation that is reached in the absence of droplet formation.

For conventional kerosene combustion, soot particles are the dominant condensation nuclei for liquid droplet activation, with subsequent freezing into ice crystals (Kärcher and Yu, 2009; Kleine et al., 2018). In addition, volatile particles mainly originating from the sulfur in the fuel may significantly contribute to ice crystal formation (Yu et al., 2024). Soot particles and sulfur are absent in the hydrogen combustion case. Therefore, other particles acting as condensation nuclei may contribute significantly to ice crystal formation. In this study, we assume that the ice crystals form on entrained ambient aerosols. To

date, it is unclear whether volatile particles originating from emitted $NO_x$ or from lubrication oil (Ungeheuer et al., 2022; Ponsonby et al., 2024; Zink et al., 2025) are abundant enough at cruise altitude to dominate ice crystal formation. Due to the higher supersaturations encountered in the hydrogen combustion case, an additional pathway to ice crystal formation could be homogeneous droplet nucleation (HDN). HDN is the process where small-droplets form out of the vapor phase without any condensation nucleus involved. For this process, high relative humidities ($\gtrsim 500\,\%$) are a prerequesite (Wölk and Strey, 2001).

In an ideal case, the exhaust air is particle-free and the HDN process is negligibile. Then, entrained ambient aerosols are the only precursors of the ice crystal formation (Bier et al., 2024).



## 1.4 Properties of ambient aerosol

Aerosol particles in the atmosphere are typically classified into four modes based on particle diameter (Petzold and Kärcher, 2012): nucleation mode (diameter $\lesssim 20\,\mathrm{nm}$), Aitken mode ($\sim$ 20-100 nm), accumulation mode ($\sim$ 100-1000 nm), and coarse mode ($\gtrsim 1\,\mu\mathrm{m}$). Nucleation mode particles form by gas-to-particle conversion, where supersaturated vapors (e.g., sulfuric acid, ammonia, nitric acid, and low-volatility organic compounds) nucleate homogeneously or on existing ions (Yu et al., 2010). Sources of these vapors in the upper troposphere (UT) include both natural and anthropogenic emissions, which are transported from lower altitudes to the UT by deep convection or large-scale atmospheric lifting (e.g., Hermann et al., 2003). Another direct source is aircraft emissions (e.g., Righi et al., 2021, 2023).

Aitken mode particles form by the growth of nucleation mode particles via condensation of vapors and coagulation of smaller particles. Some Aitken particles can also originate directly from combustion sources, such as aviation soot. Growth of Aitken particles via continued condensation and coagulation leads to the formation of accumulation mode particles. The direct contribution of aircraft emissions to the accumulation mode is typically small (Petzold et al., 1999). Unlike secondary aerosols that originate from gas-to-particle conversion, coarse mode particles are mechanically-induced primary aerosols (e.g., sea salt, dust, volcanic ash), which are transported to the UT mainly by deep convection or volcanic eruptions.

The various chemical components, ranging from well-soluble components like sulfuric acid or inorganic salts to weakly/insoluble particles like freshly emitted soot, leads to a wide range of aerosol particle solubilites (Kaiser et al., 2019). The solubility of an aerosol particle depends on its mixing state (Riemer et al., 2019). Both modeled and observed total aerosol number concentrations in the UT exhibit strong seasonal and geographic variability (Petzold et al., 2002; Minikin et al., 2003; Hermann et al., 2003; Borrmann et al., 2010; Kaiser et al., 2019; Brock et al., 2021). These studies report nucleation and Aitken mode particle number concentrations in the UT ranging from $\sim 10^2\,\mathrm{cm}^{-3}$ to $\sim 10^4\,\mathrm{cm}^{-3}$. Number concentrations of accumulation mode particles are typically lower, ranging from $\sim 10^1\,\mathrm{cm}^{-3}$ to $\lesssim 10^3\,\mathrm{cm}^{-3}$. Due to the larger size, however, the accumulation mode typically contains more mass than the nucleation and Aitken modes. Except in extreme scenarios, such as volcanic eruptions, coarse mode particle concentrations in the UT are generally several orders of magnitude lower, with number concentrations typically well below $1\,\mathrm{cm}^{-3}$ (Kaiser et al., 2019; Beer et al., 2020; Brock et al., 2021).

The strong influence of anthropogenic emissions on aerosol number concentrations in the UT introduces additional variability in future projections, as a potential reduction in emissions could result in a cleaner atmosphere. Such a reduced aerosol load was observed during the BLUESKY campaign during the COVID-19 lockdown (Voigt et al., 2022).

## 1.5 Scope and outline of the study

Our overarching goal is to develop a parameterization for the final number of ice crystals formed for aircraft with hydrogen combustion, suitable for implementation in GCMs and other large-scale contrail models. Specifically, we seek a functional relationship of the form

$$N_{\mathrm{ice,f}} = N_{\mathrm{ice,f}}(\mathbf{a}_{\mathrm{atmosphere}}, \mathbf{a}_{\mathrm{aerosol}}, \mathbf{a}_{\mathrm{aircraft}}) , \tag{2}$$





where $\mathbf{a}_{\mathrm{atmosphere}}$ denotes a set of parameters characterizing the background atmosphere (e.g., ambient temperature), $\mathbf{a}_{\mathrm{aerosol}}$
represents properties of ambient aerosols (number concentration, size distribution and solubility) and $\mathbf{a}_{\mathrm{aircraft}}$ includes aircraft-related parameters (e.g., engine size).

Our objective is to identify a functional relationship that balances simplicity with physical fidelity. In other words, we aim to capture the impact of the key processes that govern ice crystal formation in a form that is simple enough to be easily implemented in other models. In Parts 1 and 2 of a trilogy of papers, we systematically explore a broad parameter space and
address aspects that have not been explored before to gain a deep insight into the physical processes influencing the ice crystal formation. This allows us to select the set of parameters that constitute the inputs to the $N_{\mathrm{ice,f}}$ parametrization.

While a previous $N_{\mathrm{ice,f}}$ parametrization of contrail formation (Kärcher et al., 2015) used a first-principles-based concept, the current work across Parts 1 to 3 employs a hybrid approach. It combines a data-driven advanced regression method to fit a multidimensional database of contrail formation simulations with analytical scaling relations. To keep the number of
dimensions in the simulation database as small as possible, we identify conditions under which the sensitivity to specific parameters is negligible, and also employ analytical scaling relations derived from sensitivity simulations.

In Part 1, we focus on the roles of atmospheric parameters $\mathbf{a}_{\mathrm{atmosphere}}$ and aerosol parameters $\mathbf{a}_{\mathrm{aerosol}}$. Aircraft-related parameters $\mathbf{a}_{\mathrm{aircraft}}$ are addressed in Part 2. The final parameterization, synthesizing insights from Parts 1 and 2, will be presented in Part 3.

In the current study, we first provide a short review of the used contrail formation model (Sec. 2). This model is then used to investigate further (micro-)physical aspects that were not addressed in a previous publication by Bier et al. (2024). This includes the potential role of coarse mode particles in the contrail formation process (Sec. 3.1), a suitable scaling relation capturing the influence of multiple co-existing aerosol populations on the number of ice crystals (Sec. 3.2) and an investigation of the potential importance of the homogeneous droplet nucleation process (Sec. 3.3). We then present a large data set of
simulations systematically scanning the relevant parameter space (Sec. 4) before we discuss our findings in the light of the design of a $N_{\mathrm{ice,f}}$ parametrization (Sec. 5).

## 2   LCM box model

This section serves as a short review of the used model. Detailed explanations and underlying equations can be found in Bier et al. (2022, 2024).

The Lagrangian Cloud Module (LCM) was first described in Sölch and Kärcher (2010) and uses a particle-based approach (also known as super-droplet method (Shima et al., 2009; Grabowski et al., 2019)). Two variants of LCM exist: The first is fully coupled to the fluid solver EULAG and was used to simulate natural cirrus clouds (Sölch and Kärcher, 2011; Unterstrasser et al., 2017b) as well as contrails during the vortex phase (Unterstrasser and Sölch, 2010; Unterstrasser, 2014) and dispersion phase (Unterstrasser et al., 2017a). The second LCM variant is a box model version with simplified prescribed dynamics. It
was for example used to evaluate the numerical implementation and convergence of particle-based algorithms of depositional growth and nucleation (Unterstrasser and Sölch, 2014) and collisional growth (Unterstrasser et al., 2017c, 2020).





Bier et al. (2022, 2024) expanded the LCM box model version by contrail formation physics on soot and ambient aerosols, respectively. In this version, aerosol particles and hydrometeors (liquid droplets and ice crystals) are represented by simulation particles (SIPs). Each SIP represents a certain number of physical particles with identical properties. Each SIP carries information such as particle type, phase, radius, and the associated liquid or ice water mass.

In this box model version, the dilution is either analytically prescribed or obtained from a previously performed CFD simulation. The dilution governs plume expansion, cooling, and humidity evolution. The contrail microphysics is then calculated time-resolved in an offline approach without feedback on the plume dynamics. The microphysical processes considered include hygroscopic growth of aerosol particles and activation into water droplets (described by $\kappa$-Köhler theory), condensational droplet growth, homogeneous freezing of supercooled droplets, depositional ice crystal growth, and latent heat release/consumption during phase transitions.

In this study, we use the same box model version with its numerical setup as Bier et al. (2024) to simulate contrail formation on entrained ambient aerosols for hydrogen combustion. As dilution data, we use 1000 trajectories, which are a representative merged subset from the 25000 FLUDILES trajectories. These trajectories were obtained from a large-eddy simulation of the exhaust plume evolution downstream of a CFM56 engine representative for an A340-300 aircraft (Vancassel et al., 2014). Each trajectory represents a fraction of the plume volume/mass. The box model is run for each trajectory separately and results are presented as sums over all trajectories for extensive quantities (e.g., total ice crystal number) and as mass-weighted averages for intensive quantities (e.g., plume temperature).

We use the set of baseline values listed in Tab. 1 (denoted by the asterics), which are prescribed in the model setup if not stated differently. The prescribed water vapor emission index $EI_\mathrm{v}^*$ and specific combustion heat $Q^*$ for hydrogen combustion are fixed throughout the whole study. Other parameters that are not listed in Tab. 1 (ambient temperature $T_\mathrm{a}$, aerosol number concentration $n_\mathrm{aer}$) are varied in each subsection of Sec. 3 and their prescribed values are explicitly stated there.

Compared to the setup in Bier et al. (2024), we use a higher value for the overall propulsion efficiency of $\eta^* = 0.4$ (increased from 0.36), reflecting a projection towards more efficient engines. The impact of varying the overall propulsion efficiency on contrail formation is discussed in detail in Part 2 (Zink and Unterstrasser, 2025) of this paper series. Additionally, we correct a discrepancy made in Bier et al. (2024) regarding the geometric width of the log-normal aerosol size distribution. While a value of $\sigma_\mathrm{aer} = 1.6$ was reported, the actual value used in their simulations was $\sigma_\mathrm{aer} = 1.69$ (see Sec. A). Although the exact value of the geometric width has a negligible effect on the results (Fig. A1), we have corrected the bug and now use the true value of $\sigma_\mathrm{aer} = 1.6$ as reported in our setup.

## 3 Further microphysical insights

### 3.1 Impact of coarse mode particles

As written in Sec.1.4, coarse mode particles are typically several orders of magnitude fewer in number compared to coexisting nucleation, Aitken, or accumulation mode particles. Due to their low abundance, a significant direct contribution to the total ice crystal number is not expected. However, due to their large size and thus low Kelvin barrier, they can activate into water





**Table 1.** Baseline values of ambient pressure $p_a^*$, ambient relative humidity $RH_{ice,a}^*$, aircraft speed $U_\infty^*$, water vapor emission index $EI_v^*$, specific combustion heat $Q^*$, overall propulsion efficiency $\eta^*$, exit temperature $T_E^*$, exit area $A_E^*$, exit excess jet velocity $U_{jet,E}^*$ (total jet speed minus aircraft speed), geometric mean radius of aerosol particles $\overline{r}_{d,aer}^*$, geometric width $\sigma_{aer}^*$ and hygroscopicity $\kappa_{aer}^*$.

| background conditions | $p_a^* = 260\,\mathrm{hPa}$, $RH_{ice,a}^* = 115\,\%$, $U_\infty^* = 250\,\mathrm{m\,s^{-1}}$ |
|---|---|
| fuel/engine properties | $EI_v^* = 8.94\,\mathrm{kg\,kg^{-1}}$, $Q^* = 120\,\mathrm{MJ\,kg^{-1}}$, $\eta^* = 0.4$ |
| engine exit conditions | $T_E^* = 580\,\mathrm{K}$, $A_E^* = 0.25\pi\,\mathrm{m^2}$, $U_{jet,E}^* = 230\,\mathrm{m\,s^{-1}}$ |
| aerosol properties | $\overline{r}_{d,aer}^* = 20\,\mathrm{nm}$, $\sigma_{aer}^* = 1.6$, $\kappa_{aer}^* = 0.5$ |

droplets already at lower supersaturations. This raises the question of whether their earlier activation could indirectly influence the activation behavior of smaller particles by changing the water vapor concentration within the plume and therefore the final number of ice crystals formed $N_{ice,f}$ (i.e., the number when ice crystal formation is finished).

To investigate this, we performed simulations using a bimodal size distribution that includes a coarse mode with a geometric mean radius of $\overline{r}_{d,aer,2} = 2000\,\mathrm{nm}$, and a smaller mode with geometric mean radius $\overline{r}_{d,aer,1}$ (Fig. 2). The coarse mode
is assumed to be weakly soluble (hygroscopicity $\kappa_{aer,2} = 0.05$), while the smaller mode is considered either well-soluble ($\kappa_{aer,1} = 0.5$) or weakly soluble ($\kappa_{aer,1} = 0.05$). Simulations were performed with a total aerosol number concentration of $n_{aer,tot} = n_{aer,1} + n_{aer,2} = 1000\,\mathrm{cm^{-3}}$, comparing scenarios with $n_{aer,2} = \{1, 10\}\,\mathrm{cm^{-3}}$ against those without coarse mode particles ($n_{aer,2} = 0\,\mathrm{cm^{-3}}$).

Across all scenarios, the presence of coarse mode particles does not result in a noticeable change in the ice crystal number
(Fig. 2). Due to their low relative abundance and the continuous entrainment of aerosols into the plume over time, they are not abundant enough to significantly deplete the water vapor at an early stage to hinder the smaller-sized particles in their activation. This low importance of coarse mode particles was also found for other values of $n_{aer,tot}$ and $\kappa_{aer,2}$ (not shown). Therefore, if the coarse mode particles are two or more orders of magnitude fewer in number than another particle mode –as is typically the case in the UT (see Sec. 1.4)– we can safely neglect them in the ice crystal formation process.

**3.2 Scaling relation for multiple aerosol populations**

As outlined in Sec. 1.4, the aerosol spectrum consists of multiple modes. While all considered particles are of the same type (namely, ambient aerosols), they can be grouped according to differences in their size distribution and solubility. To clearly distinguish between groups of aerosols with different combinations of these properties, we introduce the term *aerosol population*, which refers to a set of aerosol particles characterized by a specific mono-modal size distribution and solubility.
Clearly, to obtain a realistic picture of ice crystal formation on ambient aerosols, multiple aerosol populations should be considered. At the same time, this comes at the cost of increased complexity due to the larger number of parameters involved. Therefore, Bier et al. (2024) have already attempted to replicate simulations involving two co-existing aerosol populations by using a simplified approach that includes only a single population characterized by averaged properties of the two. Their results





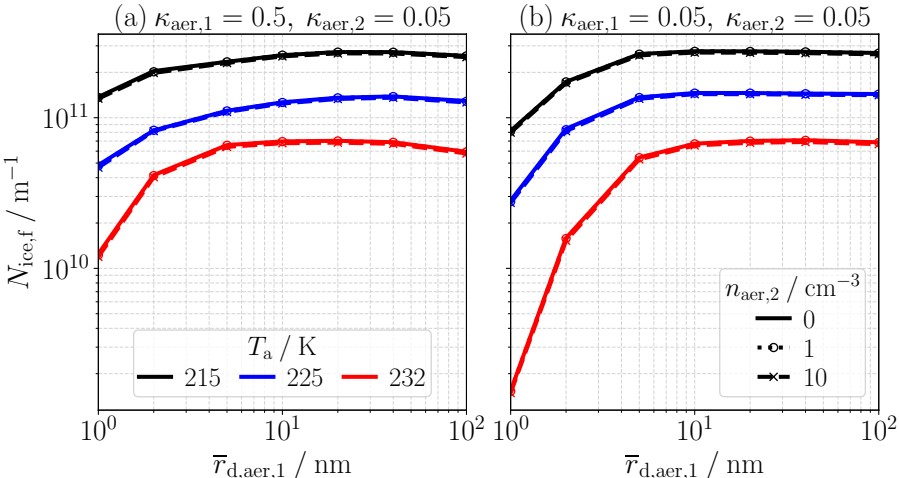

**Figure 2.** Impact of coarse mode particles with $\overline{r}_{\mathrm{d,aer,2}} = 2000\,\mathrm{nm}$ on the final number of ice crystals formed $N_{\mathrm{ice,f}}$. In all simulations, the total aerosol number concentration is $n_{\mathrm{aer,tot}} = n_{\mathrm{aer,1}} + n_{\mathrm{aer,2}} = 1000\,\mathrm{cm}^{-3}$ with varying $n_{\mathrm{aer,2}}$ values (different line styles). Shown are results as a function of the geometric radius of the smaller aerosol mode $\overline{r}_{\mathrm{d,aer,1}}$ for different ambient temperatures $T_{\mathrm{a}}$ (different colors) and different hygroscopicity combinations (panel).

demonstrated that significant discrepancies can arise between the two methods when nucleation mode particles are involved.
Motivated by these findings, we revisit this topic and introduce an alternative scaling approach in this section.

In a first step, we perform reference simulations including two co-existing populations differing in size and solubility. We prescribe a total number concentration $n_{\mathrm{aer,tot}} = 1000\,\mathrm{cm}^{-3}$ with varying ratios $n_{\mathrm{aer,1}} : n_{\mathrm{aer,2}}$. We investigate the total number of ice crystals formed for various combinations of aerosol properties and different ambient temperatures (Fig. 3). We confirm for $n_{\mathrm{aer,1}} : n_{\mathrm{aer,2}} = 1$ (as done by Bier et al. (2024)) that simulations using a single aerosol population with average properties
can reproduce those simulations well, except in cases where one geometric mean radius is smaller and the other larger than $\sim 5\,\mathrm{nm}$ (Fig. 3a-d).

Alternatively, we test a different scaling approach, where we use simulations including a single aerosol population characterized by its log-normal size distribution and solubility. Instead of the actual number concentration $n_{\mathrm{aer,i}}$ of this population, we prescribe the total aerosol number concentration $n_{\mathrm{aer,tot}}$. We then apply a weighted mean approach

$$\hat{N}_{\mathrm{ice,f}} = \sum_i \frac{n_{\mathrm{aer,i}}}{n_{\mathrm{aer,tot}}} \cdot N_{\mathrm{ice,f},i}(n_{\mathrm{aer,tot}})\,, \tag{3}$$

where $i$ goes over all aerosol populations. In Eq. (3), $\hat{N}_{\mathrm{ice,f}}$ is the approximated total ice crystal number and $N_{\mathrm{ice,f},i}(n_{\mathrm{aer,tot}})$ the simulated number of ice crystals formed on population $i$ with assumed total aerosol number concentration $n_{\mathrm{aer,tot}}$.

This weighted mean approach reasonably reproduces the reference simulations (Fig. 3). In particular, it strongly reduces the error when nucleation mode particles are involved compared to the approach using a single population with average properties.
The approach works well for equal partitions of the two number concentrations (Fig. 3a-d) as well as for unequal partitions



(Fig. 3e-f). In fact, the approach works better the more strongly the two number concentrations differ, as then the contribution of the population with lower concentration becomes less and less important (similarly to the coarse mode particles (Sec. 3.1)). For the examples shown in Fig. 3, the weighted mean approach yields a median relative error of $0.74\,\%$, with the errors ranging from a minimum of $0.003\,\%$ to a maximum of $9.95\,\%$.

From a physical point of view, the success of this weighted mean approach indicates that the total number of aerosols involved primarily drives the competition for the available water vapor. The fact that the $n_{\mathrm{aer,tot}} - n_{\mathrm{aer},i}$ particles in the reference simulation have different sizes and solubility than the aerosol population $i$, has a minor impact on how many ice particles form on aerosols belonging to population $i$, as long as the aerosols are entrained and not emitted.

So far, we restricted our analysis to two co-existing aerosol populations and to a single total number concentration of
$n_{\mathrm{aer,tot}} = 1000\,\mathrm{cm}^{-3}$. In the following, we show that the scaling approach (Eq. (3)) is generally applicable for multiple co-existing aerosol populations and different total number concentrations $n_{\mathrm{aer,tot}}$. To do so, we perform simulations incorporating in total six aerosol populations, described by three geometric mean radii ($\overline{r}_{\mathrm{d,aer}} = \{2, 20, 200\}\,\mathrm{nm}$) each combined with two hygroscopicities ($\kappa_{\mathrm{aer}} = \{0.05, 0.5\}$). We prescribe different $n_{\mathrm{aer,tot}}$ with each population assumed to contribute one sixth to the total number concentration. These simulations can be accurately reproduced using the weighted mean approach (Eq. (3))
across all investigated ambient temperatures (Fig. 4), demonstrating the broad applicability of this scaling method. For the examples shown in Fig. 4, the weighted mean approach yields a median relative error of $0.86\,\%$, with the errors ranging from a minimum of $0.08\,\%$ to a maximum of $3.65\,\%$.

Practically, the success of the weighted mean approach means that we only have to suitably represent the dependency of $N_{\mathrm{ice,f}}$ on the properties of a single population. The effect of the co-existence of multiple populations can then be emulated by
the weighted mean approach. With that, we avoid the curse of dimensionality.

### 3.3 Importance of homogeneous droplet nucleation

As described in Sec. 1.3, homogeneous droplet nucleation (HDN) with subsequent freezing into ice crystals could be another contrail ice formation pathway. For this process, high supersaturations are needed. In the hydrogen combustion case, high plume relative humidities are encountered at low ambient temperatures, with increasing peak values with decreasing ambient
temperatures (Figure 3b in Bier et al. (2024)) hinting towards a potential importance of the HDN process at low ambient temperatures.

Here, we try to assess the relative importance of the HDN process using an offline approach. To do so, we first evaluate the time-dependent nucleation rate on entrained ambient aerosols from the box model simulations via

$$J_{\mathrm{aer}}(t) = \frac{N_{\mathrm{act}}(t) - N_{\mathrm{act}}(t - \delta t)}{\delta t \cdot \hat{A}(t)} \tag{4}$$

with the box model time step $\delta t$, the effective plume area $\hat{A}$ and the number of already activated aerosols per meter of flightpath

$$N_{\mathrm{act}}(t) = N_{\mathrm{drop}}(t) + N_{\mathrm{ice}}(t) \tag{5}$$





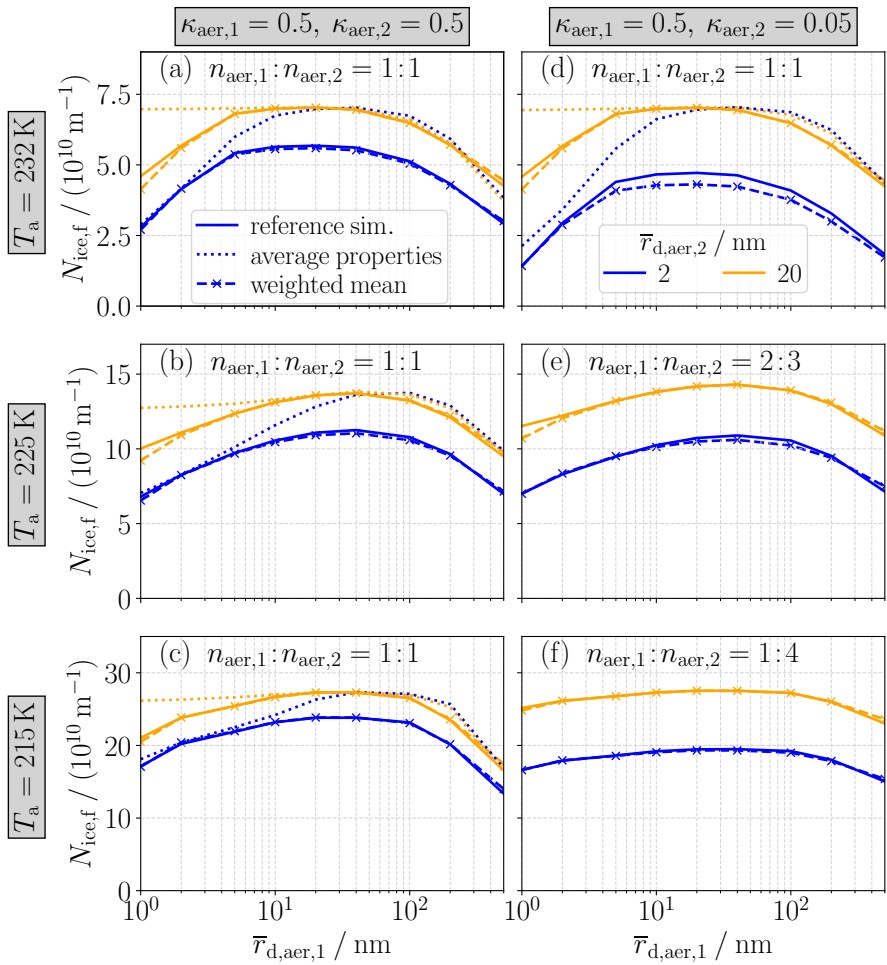

**Figure 3.** Weighted mean approach for two ambient aerosol populations. Shown is the total number of ice crystals formed $N_{\mathrm{ice,f}}$ as a function of the geometric mean radius of the first population $\overline{r}_{\mathrm{d,aer,1}}$. The color of the curves indicates the geometric mean radius of the second population $\overline{r}_{\mathrm{d,aer,2}}$. Each row corresponds to a different ambient temperature $T_{\mathrm{a}}$ and the two columns are for two different combinations of hygroscopicities $\kappa_{\mathrm{aer,1}}$ and $\kappa_{\mathrm{aer,2}}$. Results are shown for a total aerosol number concentration $n_{\mathrm{aer,tot}} = n_{\mathrm{aer,1}} + n_{\mathrm{aer,2}} = 1000\,\mathrm{cm}^{-3}$, the ratio of the two concentrations is displayed in each panel. Reference simulations including both aerosol populations in a single simulation (solid lines) are compared to the weighted mean approach after Eq. (3) (dashed lines with marker). For concentration ratios of 1:1, results for simulations including a single aerosol population with average geometric mean radius and average hygroscopicity are displayed additionally (dotted lines).





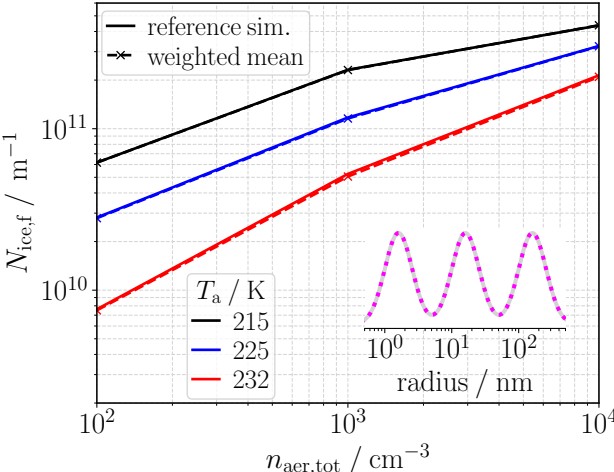

**Figure 4.** Weighted mean approach for six ambient aerosol populations. Displayed is the final number of ice crystals formed $N_{\mathrm{ice,f}}$ as a function of total aerosol number concentration $n_{\mathrm{aer,tot}}$ for different ambient temperatures $T_{\mathrm{a}}$. The six populations comprise three geometric mean radii $\overline{r}_{\mathrm{d,aer}} = \{2, 20, 200\}\,\mathrm{nm}$ each combined with two hygroscopicities $\kappa_{\mathrm{aer}} = \{0.05, 0.5\}$ (illustrated by the inset; the solid grey and the dotted magenta lines indicate the two hygroscopicities). Each population is assumed to contribute one sixth to the total aerosol number concentration. Reference simulations including all six aerosol populations in a single simulation (solid lines) are compared to the weighted mean approach after Eq. (3) (dashed lines with marker).

as sum of droplet number $N_{\mathrm{drop}}$ and ice particle number $N_{\mathrm{ice}}$. We compare this nucleation rate $J_{\mathrm{aer}}$ to potential homogeneous nucleation rates. For this we use the empirical nucleation rate (Wölk and Strey, 2001; Wölk et al., 2002)

$$J_{\mathrm{HDN}} = J_{\mathrm{BD}} \exp\left(A + \frac{B}{T}\right) \tag{6}$$

with the empirical constants $A = -27.56$ and $B = 6500\,\mathrm{K}$ and the classical Becker-Döring nucleation rate

$$J_{\mathrm{BD}} = \sqrt{\frac{2\sigma}{\pi m}}\, v \left(\frac{e_{\mathrm{v}}}{k_{\mathrm{B}}T}\right)^2 \exp\left\{\frac{-16\pi v^2 \sigma^3}{3 k_{\mathrm{B}} T^3 (\ln S)^2}\right\}\quad. \tag{7}$$

In Eq. (7), $\sigma$ is the surface tension of the critical cluster of water molecules, $m$ is the mass of a water molecule, $v$ is the molecular volume, $k_{\mathrm{B}}$ is the Boltzmann constant, $T$ is the temperature, and $S = e_{\mathrm{v}}/e_{\mathrm{v,sat,wat}}$ is the saturation ratio (relative humidity given as ratio and not in percentage) with the actual partial pressure of water vapor $e_{\mathrm{v}}$ and the equilibrium partial vapor pressure over a flat water surface $e_{\mathrm{v,sat,wat}}$.

In our offline approach, we evaluate $J_{\mathrm{HDN}}(t)$ via Eq. (6) by using the time series $T(t)$ and $S(t)$ as obtained from existing box model simulations. For simulations with microphysics switched off (ambient aerosol number concentration $n_{\mathrm{aer}} = 0\,\mathrm{cm}^{-3}$), the time evolution of $T$ and $S$ over time is governed solely by dilution along the mixing line. For $n_{\mathrm{aer}} > 0\,\mathrm{cm}^{-3}$, the temperature evolution differs minimally due to latent heating from phase transitions. The evolution of the saturation ratio $S$ is strongly affected by the choice of $n_{\mathrm{aer}}$ value due to the water vapor depletion by activated aerosols (Fig. 5a-b).



Since our offline approach does not consider water vapor consumption by homogeneously nucleated droplets, it tends to overestimate $J_{\mathrm{HDN}}$ at later times $t$. Furthermore, it does not determine whether clusters formed at a given time are stable and grow or whether they evaporate thereafter. However, as this approach tends to overestimate $J_{\mathrm{HDN}}$, it allows for a conservative estimate of the importance of the HDN process.

At the ambient temperature $T_{\mathrm{a}} = 205\,\mathrm{K}$, the encountered saturation ratios $S$ are high (Fig. 5a), and thus also the rates $J_{\mathrm{HDN}}$ for low ambient aerosol number concentrations $n_{\mathrm{aer}}$ (Fig. 5c). At these low $n_{\mathrm{aer}}$, the maximum of the calculated rates $J_{\mathrm{HDN}}$ exceeds the maximum of the rates $J_{\mathrm{aer}}$. However, when ambient aerosols are more abundant with substantial depletion of the water vapor, the calculated rates $J_{\mathrm{HDN}}$ become several orders of magnitude lower than the corresponding rates $J_{\mathrm{aer}}$ (dotted lines for $n_{\mathrm{aer}} = 1000\,\mathrm{cm}^{-3}$ in Fig. 5c).

At an higher ambient temperature ($T_{\mathrm{a}} = 210\,\mathrm{K}$), the encountered supersaturations $S$ are lower (Fig. 5b), resulting in $J_{\mathrm{HDN}}$ values that remain several orders of magnitude below the rates $J_{\mathrm{aer}}$, even for low $n_{\mathrm{aer}}$ (Fig. 5d). In addition, the maxima of $J_{\mathrm{HDN}}$ occur much later than those of $J_{\mathrm{aer}}$. Consequently, indirect effects such as a significant activation suppression of ambient aerosols due to the water vapor depletion by a few spontaneously nucleated droplets are highly unlikely.

After displaying the time evolution of nucleation rates for the two example cases (Fig. 5), we now extend our analysis to a broader set of scenarios. For these scenarios, we evaluate the maxima of the nucleation rates (Fig. 6a). The maximum of $J_{\mathrm{aer}}$ is almost independent of ambient temperature $T_{\mathrm{a}}$ and pressure $p_{\mathrm{a}}$ at low ambient temperatures and scales nearly linearly with ambient aerosol number concentration $n_{\mathrm{aer}}$ (Tab. 2). In contrast, the maximum of $J_{\mathrm{HDN}}$ depends non-linearly on the supersaturation and is therefore highly sensitive to both $T_{\mathrm{a}}$ and $p_{\mathrm{a}}$.

As the highest potential $J_{\mathrm{HDN}}$ values occur in the absence of ambient aerosols ($n_{\mathrm{aer}} = 0\,\mathrm{cm}^{-3}$), we use these rates for a most conservative estimate. In particular, we evaluate the maxima of the rates $J_{\mathrm{HDN},n_{\mathrm{aer}}=0}$ for various combinations of $p_{\mathrm{a}}$ and $T_{\mathrm{a}}$. Isolines of $\max(J_{\mathrm{HDN},n_{\mathrm{aer}}=0})$ are shown in Fig. 6b. Since the maxima of $J_{\mathrm{aer}}$ are at least one order of magnitude above $10^{6}\,\mathrm{m}^{-3}\,\mathrm{s}^{-1}$ for aerosol number concentrations $n_{\mathrm{aer}} \geq 1\mathrm{cm}^{-3}$ (Tab. 2), we choose the isoline $\max(J_{\mathrm{HDN},n_{\mathrm{aer}}=0}) = 10^{6}\,\mathrm{m}^{-3}\,\mathrm{s}^{-1}$ as a conservative boundary for an estimation where HDN may be important. This boundary can be approximated by

$$p_{\mathrm{a}}(T_{\mathrm{a}}) = a \cdot T_{\mathrm{a}}^{2} + b \cdot T_{\mathrm{a}} + c \tag{8}$$

with $a = 1.04 \cdot 10^{2}\,\mathrm{Pa/K^{2}}$, $b = -4.14 \cdot 10^{4}\,\mathrm{Pa/K}$, and $c = 4.12 \cdot 10^{6}\,\mathrm{Pa}$.

If a $p_{\mathrm{a}}$–$T_{\mathrm{a}}$ pair lies to the right/below of this boundary (white region in Fig. 6b), we can be very sure that HDN does not play a significant role as the nucleation rates on the ambient aerosols are substantially higher regardless of the ambient aerosol number concentration. In the gray region of Fig. 6b, it remains uncertain whether HDN may significantly contribute to droplet formation. This depends on the aerosol number concentration and requires an explicit, time-resolved simulation of relative humidity and temperature evolution, along with the kinetics of droplet cluster growth/evaporation.

In the following, we argue why Eq. (8) remains applicable even when the prescribed overall propulsion efficiency $\eta$ differs from the baseline value $\eta^{*} = 0.4$. Both the overall propulsion efficiency $\eta$ and the ambient pressure $p_{\mathrm{a}}$ influence the slope of the mixing line (Eq.(1)), and the ratio $p_{\mathrm{a}}/(1-\eta)$ controls the maximum supersaturation reached for a given ambient temperature





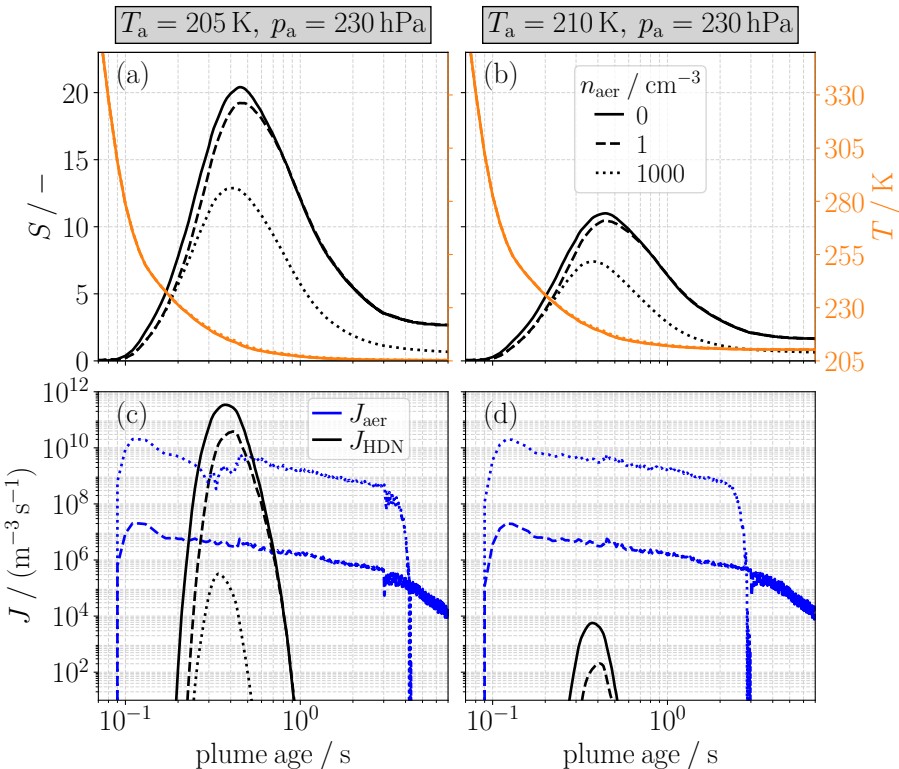

**Figure 5.** (a)-(b) Time evolution of saturation ratio $S$ (black curves) and plume temperature $T$ (orange curves) for different ambient aerosol number concentrations (indicated by the linestyle). (c)-(d) Nucleation rates on ambient aerosols $J_{\mathrm{aer}}$ after Eq. (4) (blue curves) and homogeneous droplet nucleation rates $J_{\mathrm{HDN}}$ obtained by inserting the time evolutions $S$ and $T$ into Eq. (6) (black curves). The two columns are for two different ambient temperatures $T_{\mathrm{a}}$ for the same ambient pressure $p_{\mathrm{a}}$.

$T_{\mathrm{a}}$ and ambient relative humidity $RH_{\mathrm{ice,a}}$. Therefore, instead of the actual pressure $p_{\mathrm{a}}$, the adjusted pressure

$$\tilde{p}_{\mathrm{a}} = p_{\mathrm{a}} \frac{1-\eta^*}{1-\eta} \tag{9}$$

has to be used to determine the corresponding region in Fig. 6b when the prescribed $\eta$ differs from the baseline value $\eta^* = 0.4$. Moreover, Eq. (8) can also be applied when the ambient relative humidity deviates from the baseline value $RH^*_{\mathrm{ice,a}} = 115\%$,

as such a deviation only causes a slight shift of the mixing line and has compared to a change in $T_{\mathrm{a}}$ and $p_{\mathrm{a}}$ only a minor effect on the maximum supersaturation.

**Table 2.** Maximum of nucleation rate on ambient aerosols for different number concentrations.

| $n_{\mathrm{aer}} / \mathrm{cm}^{-3}$ | 1 | 10 | 100 | 1000 |
|---|---|---|---|---|
| $\max(J_{\mathrm{aer}}) / (\mathrm{m}^{-3}\mathrm{s}^{-1})$ | $> 10^7$ | $> 10^8$ | $> 10^9$ | $> 10^{10}$ |





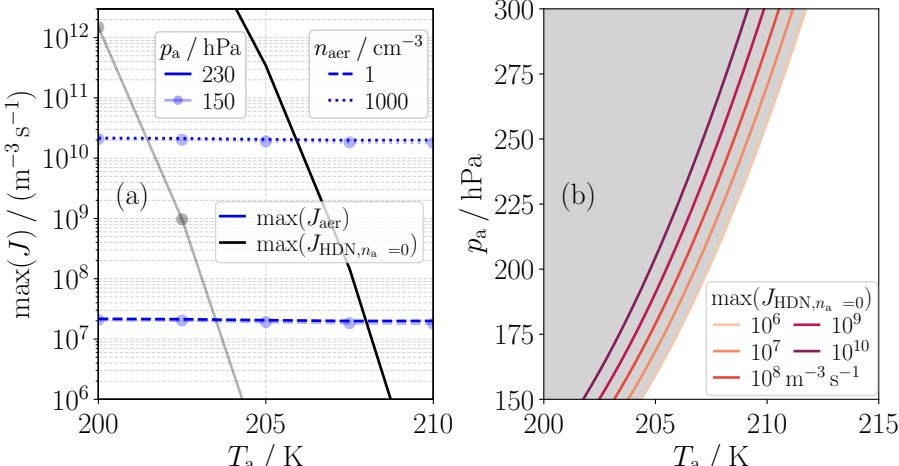

**Figure 6.** (a) Maxima of nucleation rates $\max(J)$ as a function of ambient temperature $T_a$ for two ambient pressures $p_a$ (indicated by transparency and dotted marker). Depicted are the maximum nucleation rates connected with ambient aerosols $\max(J_{aer})$ (blue lines) for two aerosol number concentrations $n_{aer}$ (different linestyle) and the maximum rates of homogeneous droplet nucleation obtained when moving along the mixing line $\max(J_{HDN,n_{aer}=0})$ (black lines). (b) Isolines for maxima of homogeneous droplet nucleation rates obtained when moving along the mixing line for different $T_a$ and $p_a$ combinations.

## 4 Systematic investigation

### 4.1 Simulation data base

Bier et al. (2024) primarily used a one- or two-at-a-time approach to investigate the main sensitivities of the number of ice
crystals formed $N_{ice,f}$ to different parameters. This method involves starting from a set of baseline values and varying one or two parameters at a time while keeping all others fixed. The goal of this section is to explore the parameter space more systematically. First, we expand the range of parameter values beyond those considered by Bier et al. (2024). Second, instead of varying one or two parameters at a time, we perform simulations for all possible combinations of parameter values (schematically illustrated in Fig. 7 for two parameters). This approach results in a total of $n_{sim} = \prod_i n_i$ simulations, where $n_i$ denotes
the number of values explored for parameter $i$. For the parameter values listed in Tab. 3, this leads to more than $n_{sim} > 20,000$ simulations. This systematic investigation enables us to make general statements about the sensitivity of $N_{ice,f}$ to different parameters across various regions of the parameter space. Moreover, the resulting comprehensive dataset provides a suitable basis for a data-driven parameterization approach. Alternative sampling strategies, such as Latin hypercube sampling, could reduce the number of simulations required. However, since the computational cost of the box model is low, we can afford to
use the full-factorial sampling strategy.

Ambient temperatures are specified over a broad range, prescribed linearly in 5 K steps from $T_a = 210\,K$ to $T_a = 230\,K$. We use a finer resolution for $T_a \geq 230\,K$ to capture the non-linear dependency on the homogeneous freezing process in this





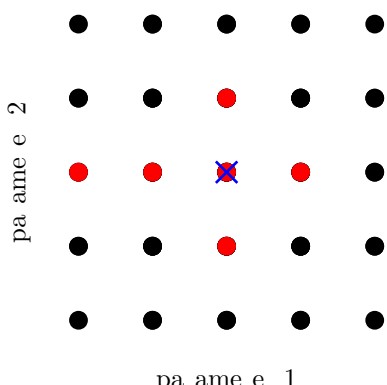

**Figure 7.** Schematic depiction of the difference between a one-at-a-time approach and the simulation setup used in the current study. Starting from a set of baseline values (blue cross), the values of a single parameter are varied in a one-at-a-time approach while all other parameters are kept constant (red dots). In the current study, we set up simulations for all possible combinations of parameter values (black dots).

temperature range. As an upper limit for the homogeneous freezing temperature, we set $235\,\mathrm{K}$. Above $T_\mathrm{a} \geq 235\,\mathrm{K}$, we assume that supercooled droplets do not freeze into ice crystals.

The range of prescribed ambient pressures reflects a wide altitude range for possible aircraft traffic. Clearly, as we performed simulations for all possible combinations of ambient temperature and pressure, highly unlikely atmospheric scenarios are also part of the dataset. Nonetheless, ambient pressure primarily influences supersaturation levels at a given ambient temperature through its effect on the slope of the mixing line (Eq. (1)). Thus, even the less likely combinations represent extrapolations that still capture the primary influence of ambient pressure on the contrail formation process.

We only consider ice (super-)saturated conditions ($RH_\mathrm{ice,a} \geq 100\,\%$) as only contrails formed under these conditions will persist for a longer time. For the chosen upper limit ($RH_\mathrm{ice,a} = 140\,\%$), the environment is sub-saturated with respect to water for all investigated ambient temperatures (e.g., $RH_\mathrm{ice,a} = 140\,\%$ corresponds to $RH_\mathrm{wat,a} \approx 80\,\%$ at $T_\mathrm{a} = 210\,\mathrm{K}$ and to $RH_\mathrm{wat,a} \approx 96\,\%$ at $T_\mathrm{a} = 234.5\,\mathrm{K}$). If the environment were supersaturated with respect to water, natural cirrus clouds would be anyway present (Koop et al., 2000). Even under water sub-saturated conditions, aerosols may grow hygroscopically large

enough to freeze and initiate natural cirrus formation in some scenarios. However, this study does not address contrail formation in the presence of existing natural cirrus, and we therefore assume a cloud-free background in all simulations.

As the impact of multiple co-existing aerosol populations can be inferred from individual simulations, each including only one of the populations with a suitably adjusted number concentration (Sec. 3.2). Hence, it is sufficient to explore the dependency





on the properties of single aerosol populations systematically. The prescribed aerosol number concentrations span several
orders of magnitude, representing the large variability in the atmosphere. As the number of ice crystals formed is only weakly
dependent on the geometric width (Fig. A1), this parameter is held constant at $\sigma_{\mathrm{aer}} = 1.6$ across all simulations (Brock et al.,
2021; Bier et al., 2024). In contrast, the geometric mean radius is varied using an approximately logarithmic spacing, covering
the full range from small nucleation mode particles to large accumulation mode particles. Based on the findings in Sec. 3.1, we
do not cover coarse mode particles. The nearly logarithmic spacing provides enhanced resolution in linear space for the small-
sized particles to capture the non-linear influence of the Kelvin effect on the droplet activation. Finally, prescribed aerosol
hygroscopicity values span several orders of magnitude to account for the diverse solubilities of potential aerosol species (see
Tab. 1 in Petters and Kreidenweis, 2007).

**Table 3.** Box model simulations for all possible combinations of listed parameter values are performed.

| parameter | values |
|---|---|
| ambient temperature $T_{\mathrm{a}}$ / K | 210, 215, 220, 225, 230, 232.5, 234.5 |
| ambient pressure $p_{\mathrm{a}}$ / hPa | 150, 230, 320, 400 |
| ambient relative humidity over ice $RH_{\mathrm{ice, a}}$ / % | 100, 115, 125, 140 |
| aerosol number concentration $n_{\mathrm{aer}}$ / cm$^{-3}$ | 1, 10, 100, 1000, 10000 |
| aerosol geometric mean radius $\overline{r}_{\mathrm{d, aer}}$ / nm | 1, 2, 5, 10, 20, 40, 100, 200, 500 |
| aerosol hygroscopicity $\kappa_{\mathrm{aer}}$ / − | 0.01, 0.05, 0.25, 1.25 |

## 4.2 Systematic evaluation of the ice crystal number

In this section, we present the results of the $n_{\mathrm{sim}} > 20{,}000$ simulations in a condensed manner. As a first step, we investigate
the mean dependency of the final number of ice crystals formed $N_{\mathrm{ice,f}}$ on the various parameters. To do so, we look at all
simulations that use a specific value $k$ of the parameter $i$, and calculate the arithmetic mean of $N_{\mathrm{ice,f}}$ across these $n_{\mathrm{sim}}/n_i$
simulations, where $n_i$ is the number of values considered for parameter $i$.

This analysis confirms several findings already reported by Bier et al. (2024). The mean number of ice crystals decreases with
increasing ambient temperature $T_{\mathrm{a}}$ (Fig. 8a), owing to lower and shorter-lasting supersaturation at higher temperatures. Above
$T_{\mathrm{a}} \sim 230\,\mathrm{K}$, a pronounced nonlinear decline in ice crystal number is observed, attributed to the large freezing time scales of
supercooled droplets at those temperatures. Unlike conventional contrail formation on soot particles, where ice crystal numbers
saturate at low ambient temperatures, our results show a continuous increase in ice crystal number with decreasing $T_{\mathrm{a}}$ due to
continuous entrainment of ambient aerosols into the plume and prolonged supersaturation at lower temperatures.

Higher ambient pressure $p_{\mathrm{a}}$ results in increased ice crystal numbers (Fig. 8b), driven by enhanced supersaturation caused by
a steeper slope of the mixing line (Eq. (1)). The mean ice crystal number shows only a weak dependence on ambient relative
humidity, with a slight increase with increasing $RH_{\mathrm{ice,a}}$ (Fig. 8c).




The mean number of ice crystals formed strongly depends on the background aerosol number concentration $n_{\mathrm{aer}}$ (Fig. 8d). At low $n_{\mathrm{aer}}$, the relationship is approximately linear, whereas saturation effects become apparent for high $n_{\mathrm{aer}}$. For $\bar{r}_{\mathrm{d,aer}} \lesssim 10\,\mathrm{nm}$, the mean ice crystal number increases with increasing geometric mean radius (Fig. 8e), driven by the nonlinear Kelvin effect. Between $\bar{r}_{\mathrm{d,aer}} \gtrsim 10\,\mathrm{nm}$ and $\bar{r}_{\mathrm{d,aer}} \lesssim 100\,\mathrm{nm}$, the dependency is weak. Since Bier et al. (2024) only investigated values up to $\bar{r}_{\mathrm{d,aer}} = 70\,\mathrm{nm}$, they did not report the decrease in ice crystal number observed for $\bar{r}_{\mathrm{d,aer}} \gtrsim 100\,\mathrm{nm}$. This decline is due to the freezing-point depression of solution droplets, which will be discussed in more detail further below. In this mean-based analysis, the aerosol hygroscopicity $\kappa_{\mathrm{aer}}$ has only a minor impact on the ice crystal number (Fig. 8e).

To gain deeper insight, we refine our analysis by still considering a specific value $k$ of parameter $i$, while additionally grouping the data by the $n_j$ values of a co-parameter $j$. Within each subgroup, we compute the arithmetic mean over the $n_{\mathrm{sim}}/(n_i \cdot n_j)$ simulations and plot the obtained $n_j$ mean values as colored dots in Fig. 8. The spread of the dots indicates how sensitive the ice crystal number is to the considered co-parameter at a given parameter value $k$.

This analysis confirms that ambient aerosol number concentration and ambient temperature are the most influential parameters throughout the examined parameter space. Furthermore, the extent of the dots reveals that the ice crystal number is significantly sensitive to $RH_{\mathrm{ice,a}}$ only when $T_{\mathrm{a}} \gtrsim 230\,\mathrm{K}$ (Fig. 8a). Additionally, this analysis indicates that the hygroscopicity $\kappa_{\mathrm{aer}}$ becomes influential for $\bar{r}_{\mathrm{d,aer}} \lesssim 10\,\mathrm{nm}$ and $\bar{r}_{\mathrm{d,aer}} \gtrsim 100\,\mathrm{nm}$ (Fig. 8e). This is caused by two distinct physical processes: A higher hygroscopicity $\kappa_{\mathrm{aer}}$ reduces the critical supersaturation required for droplet activation. This becomes important when the Kelvin effect is strong, i.e., for small-sized particles. Conversely, a higher $\kappa_{\mathrm{aer}}$ value lowers the freezing temperature of supercooled droplets. While the freezing behavior of weakly soluble activated aerosols is similar to that of pure supercooled droplets, the presence of soluble material within a supercooled droplet inhibits the initiation of the freezing process (Koop et al., 2000). This means that some well-soluble aerosols activate into droplets but do not freeze. This suppression becomes more pronounced for larger dry particle radii due to the higher solute content for a given water content.

All together, this systematic investigation shows that the number of ice crystals is almost independent of ambient relative humidity, ambient aerosol size, and hygroscopicity for the parameter subspace defined by $T_{\mathrm{a}} \leq 225\,\mathrm{K}$ and $10\,\mathrm{nm} \leq \bar{r}_{\mathrm{d,aer}} \leq 100\,\mathrm{nm}$. This conclusion is supported by an analysis restricted to this subspace, as shown in Fig. 9.

## 5 Discussion and Conclusions

In this section, we summarize the key findings of our study and discuss their implications for the development of an upcoming parameterization for the number of ice crystals formed. We conducted simulations using the particle-based LCM box model to investigate contrail formation on entrained ambient aerosols for hydrogen combustion. Our focus was on microphysical processes with the main findings being:

- Coarse mode particles typically have not to be accounted for as their direct and indirect influence on the ice crystal formation is negligible due to their typical much lower number concentration compared to co-existing nucleation, Aitken, or accumulation mode particles.





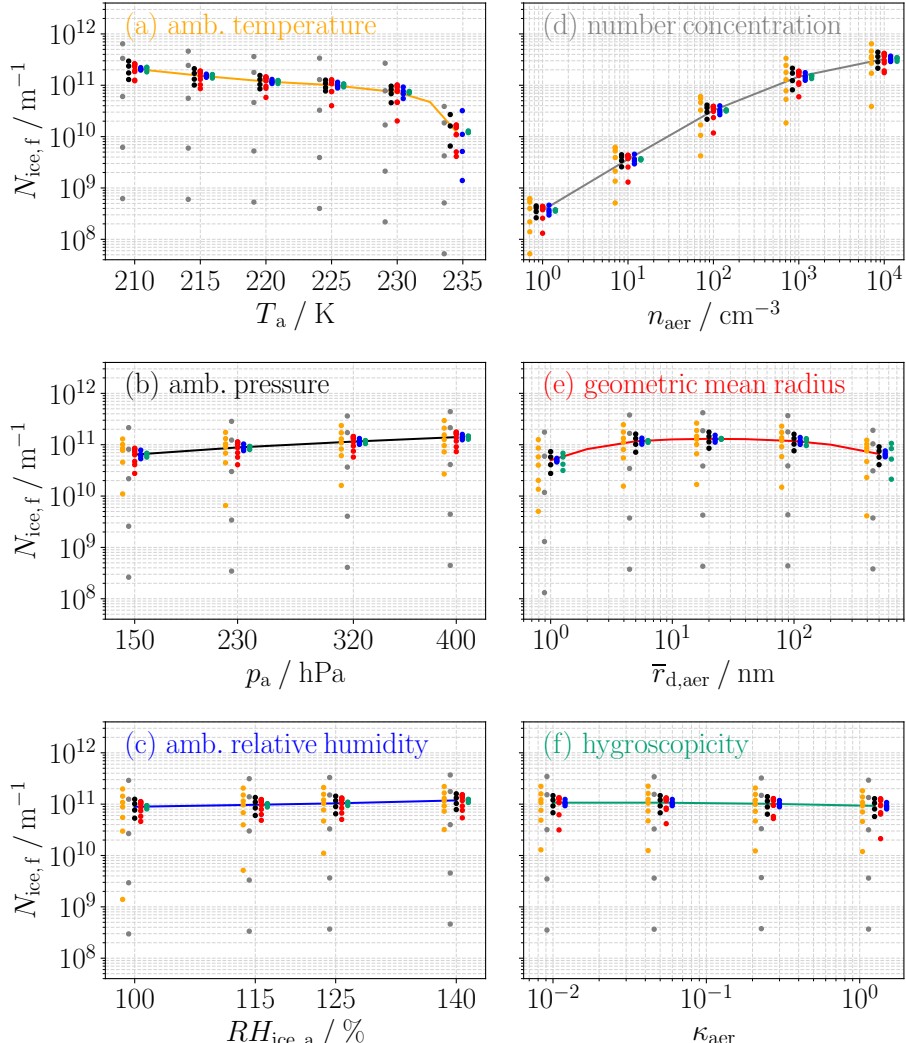

**Figure 8.** Mean dependency of the final number of ice crystals formed $N_{\mathrm{ice,f}}$ on the various parameters (different colored solid lines in each panel). The colored dots show the mean $N_{\mathrm{ice,f}}$ calculated within subgroups grouped by the specific parameter value at the x-axis and by the values of a co-parameter. The color of the dots indicates the chosen co-parameter. The dots for the different co-parameters correspond to the parameter value at the center but are shifted for visualization purposes. In (a), the dependency on the co-parameters is not displayed for $T_{\mathrm{a}} = 232.5\,\mathrm{K}$ and in (e) the dependency on the co-parameters is only displayed for each second geometric mean radius.

– We showed that we can approximate the ice crystal number formed on multiple aerosol populations with a weighted mean approach of simulations that include only a single population. Therefore, it is sufficient to suitably parametrize the dependency of the ice crystal number on the properties of a single population. The ice crystal number formed on multiple populations with different properties can then be obtained by this weighted mean approach.



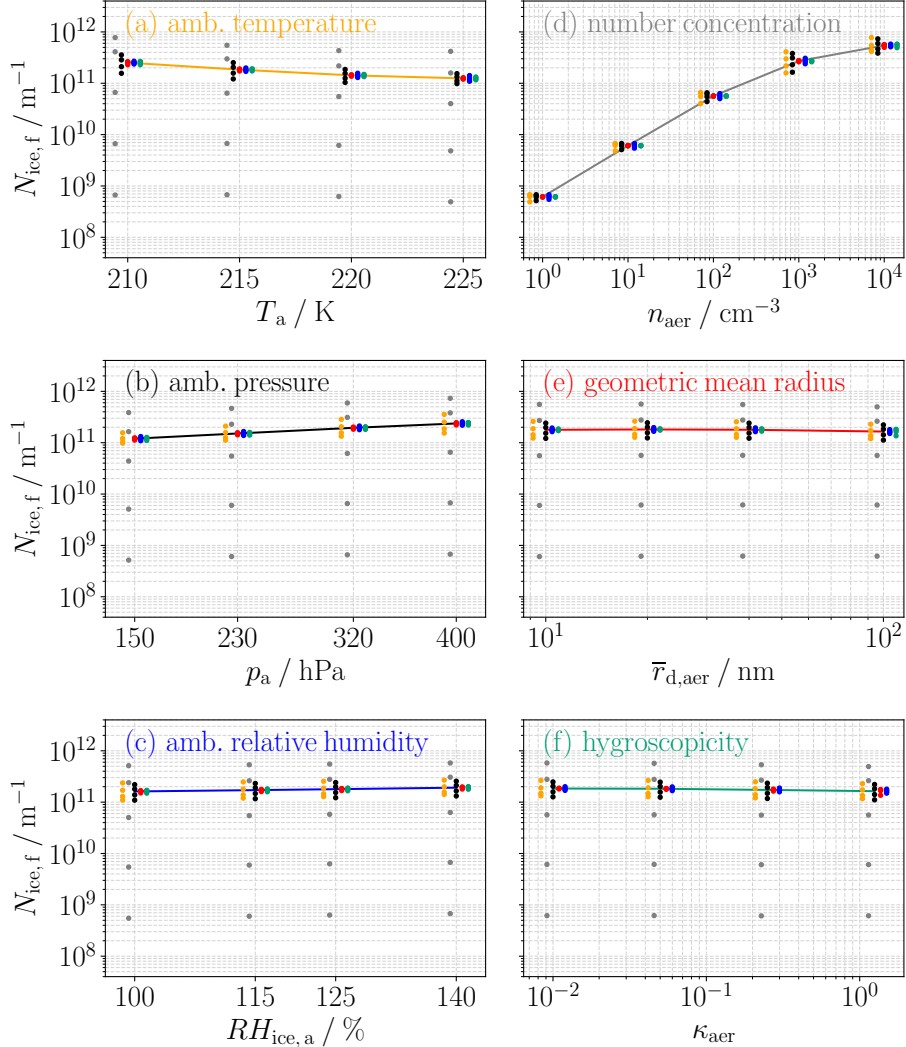

**Figure 9.** Same as Fig. 8, but for the subspace defined by $T_a \leq 225\,\mathrm{K}$ and $10\,\mathrm{nm} \leq \overline{r}_{d,\,aer} \leq 100\,\mathrm{nm}$.

– We provided a formula for a conservative boundary dividing the parameter space into regions where spontaneous nucle-
ation can safely be neglected and where this process might play a role in ice crystal formation. For a parametrization
based on contrail formation on ambient aerosol particles, the boundary can serve as a first indication of whether neglect-
ing spontaneous nucleation is justified for a specific use case.

– A systematic investigation confirmed that ambient temperature and aerosol number concentration are the most important
predictors for the number of ice crystals formed. Moreover we showed that for the subspace defined by $T_a \leq 225\,\mathrm{K}$
and $10\,\mathrm{nm} \leq \overline{r}_{d,\,aer} \leq 100\,\mathrm{nm}$, the final number of ice crystals formed becomes nearly independent of ambient rela-




tive humidity as well as the size and solubility of ambient aerosols. This finding enables the possibility to develop a parametrization with reduced complexity within this sub-regime.

In Part 2 of this paper series (Zink and Unterstrasser, 2025), we present findings that focus on engine-related aspects rather than microphysical processes in the contrail formation process. Finally, in the forthcoming Part 3, we will introduce a parameterization based on a neural network that has been trained with the full set of simulation results. It is designed to meet the

constraints and practical needs for implementing in a general circulation model or other large-scale contrail models. These models can then be used to estimate the radiative impact of contrails from a fleet of aircraft with hydrogen combustion.

*Data availability.* The presented data are available from the corresponding author upon request (josef.zink@dlr.de)

## Appendix A

The lognormal particle-size distribution reads

$$n(r) = \frac{N}{\sqrt{2\pi}} \frac{1}{\ln \sigma_r} \frac{1}{r} \exp\left[-\frac{1}{2}\left(\frac{\ln r/\overline{r}}{\ln \sigma_r}\right)^2\right] , \tag{A1}$$

where $r$ is the particle radius, $n(r)$ the number density, $N$ the total number, $\overline{r}$ the geometric mean radius and $\sigma_r > 1$ the geometric width. The geometric width $\sigma_m$ of a corresponding mass distribution is related to $\sigma_r$ by

$$\sigma_r = \sigma_m^k , \tag{A2}$$

where $k = 1/3$ for spherical particles. In the box model, $\sigma_m$ is used internally. By mistake, Bier et al. (2024) used the relation

$\sigma_{r,\text{wrong}} = k \cdot \sigma_m$ instead of Eq. (A2), resulting in incorrect values shown in their Fig. 5d. The correct values $\sigma_{r,\text{correct}}$ are then given by

$$\sigma_{r,\text{correct}} = (3 \cdot \sigma_{r,\text{wrong}})^{1/3} . \tag{A3}$$

These corrected values are displayed in Fig. A1.

*Author contributions.* **Josef Zink**: Conceptualization, Data curation, Formal analysis, Investigation, Methodology, Software, Validation,

Visualization, Writing – original draft, Writing – review & editing, **Simon Unterstrasser**: Conceptualization, Methodology, Funding acquisition, Software, Supervision, Project Administration, Writing – review & editing **Ulrike Burkhardt**: Discussion

*Competing interests.* The contact author has declared that none of the authors has any competing interests.



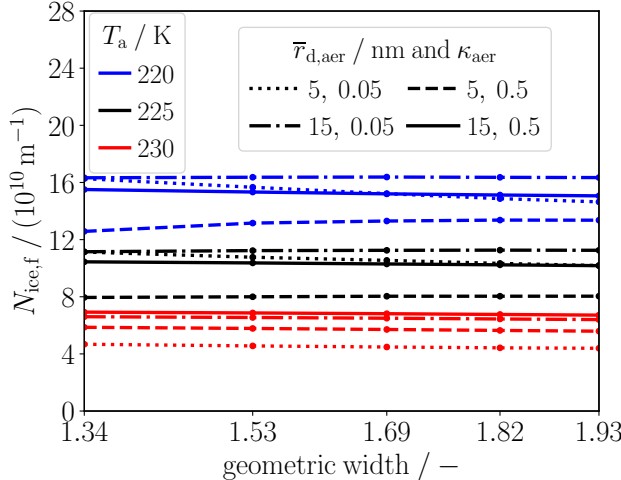

**Figure A1.** Corrected Fig. 5d of Bier et al. (2024). Depicted is the final number of ice crystals formed $N_{ice,f}$ as a function of the geometric width of the background aerosol distribution. Simulation results are shown for different ambient temperatures $T_a$ and different combinations of geometric mean radius $\overline{r}_{d,aer}$ and hygroscopicity $\kappa_{aer}$ for the number concentration $n_{aer} = 600\,\mathrm{cm}^{-3}$. In the original figure, wrong values of the geometric width are displayed on the x-axis.

*Acknowledgements.* This work has been funded by Airbus and by the DLR internal project "H2CONTRAIL". We thank X. Vancassel
(ONERA) for providing CFD data on plume dilution. We thank M. Righi for an internal review of the paper draft. Furthermore, we thank C. Weiß-Rehm, and C. Renard for their comments. The language was polished with the help of ChatGPT.





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
