# Peer review of "Contrail formation for aircraft with hydrogen combustion - Part 1: A systematic microphysical investigation"

_EGUsphere, 2025_

## Referee Comment (RC2)

**Review Report for ACP**

**Contrail formation for aircraft with hydrogen combustion – Part 1: A systematic microphysical investigation**
**Zink, Unterstrasser & Burkhardt (EGUsphere preprint, 2025)**

The authors perform a comprehensive suite of Lagrangian Cloud Model (LCM) simulations to study ice nucleation on ambient aerosols in hydrogen combustion plumes. They quantify the negligible influence of coarse-mode aerosols, propose a weighted-mean scaling to approximate multi-population behavior, and derive a conservative boundary (Eq. 8) to determine when HDN can be neglected. Furthermore, they identify a parameter subspace (temperature $\leq$ 225 K, 10–100 nm aerosol size) where ice number formation ($N_{\text{ice},f}$) becomes insensitive to aerosol properties, thus facilitating parametrization.

**General comments**

The manuscript is scientifically valuable, presenting a thorough microphysical modelling study of contrail formation for hydrogen combustion and explores important conceptual simplifications for parametrization development. However, issues regarding the structure, novelty, model validation, assumptions, and clarity of applicability need to be addressed before publication.

Specifically, the work is well executed, carefully documented, and useful for the contrail/aviation community. However, it mostly reports comprehensive application of an existing particle-based Lagrangian Cloud Module and a large parameter sweep; it does not introduce a clear, novel physical mechanism in atmospheric chemistry or cloud microphysics. As such, in its present form it reads as an important and careful modeling/data paper but falls short of the level of novel atmospheric physics/chemistry unless the authors: (a) clarify and amplify the particular microphysical insight(s) that are genuinely new; (b) add further validation/uncertainty quantification; and (c) improve the argument that the identified insensitive subspace is a new physical result rather than an empirical property of this model/setup. As a consequence, the authors should note that splitting the research into 3 manuscripts is acceptable only if each part contains a clear, stand-alone novelty claim. As written, Part 1 is a thorough model-based analysis, and may be regarded as insufficiently novel for ACP on its own.

We elaborate this general comment with further specific comments below:

**Specific comments** [6pt]

1. Scope and Contribution: While the manuscript is scientifically valuable and the discussion of results is thorough, the novelty appears circumscribed to systematically running an existing model (Bier et al., 2024). It feels more like a follow-up study rather than a substantial, standalone modelling contribution. The authors mention splitting the analysis into three parts. I am unconvinced that the current contribution is sufficient to justify a trilogy of papers. I strongly suggest the authors consider compressing the work into one or two meaningful papers. For instance, the third part (building an AI-based surrogate model) seems to offer the distinct modelling contribution that is currently lacking in this first part; integrating these findings could significantly strengthen the publication.

2. Introduction and Motivation: The introduction is currently too long and fragmented. The critique of the state-of-the-art is not sufficiently connected to the specific scientific contributions of this study. Furthermore, the motivation needs to be more sharply focused on the aviation industry's context.

3. Outdated Context: The reliance on the Airbus (2020) reference for motivation feels outdated. In 2020, the target for service entry was around 2035. However, as of 2025, timelines have shifted towards 2040–2045 due to infrastructure challenges (e.g., lack of hydrogen infrastructure at airports). Indeed, I have the feeling that the H$_2$ aircraft is not a priority anymore... In any case, please, try to reflect the current industrial reality to ensure the motivation is robust and up-to-date.

4. The study is purely model-based, with no quantitative validation. Although hydrogen contrails are not yet observed, consistency checks with kerosene contrails or known ranges of $N_{\mathrm{ice},f}$ would be recommended. Alternatively, explicitly state the limits of observational validation for hydrogen combustion and discuss implications for model uncertainty.

5. Include sensitivity runs or describe variability when other subsets or entrainment efficiencies are used (now, the entrainment efficiency seems to be fixed in the paper).

6. Criterion for HDN relevance (Eq. 8) should be further discussed. The conservative criterion ($J_{\mathrm{HDN}}, n_{\mathrm{aer}} = 0 = 10^6\,\mathrm{m}^{-3}\mathrm{s}^{-1}$) is useful but requires clearer justification. Specifically, it is recommended to explain the physical rationale for this threshold and demonstrate how outcomes vary if it changes by an order of magnitude.

7. Further discuss/quantify how neglecting vapor depletion or droplet interactions may overestimate HDN frequency. The exclusion of lubrication-oil or NOx-product aerosols may limit generality. In this respect, it is recommended to discuss how their presence could modify the current conclusions, especially the negligible coarse-mode effect.

8. The paper neglects the coarse-mode particles without quantified depletion analysis. The manuscript asserts these particles can be ignored when they are orders of magnitude less numerous, but provides no order-of-magnitude vapor budget demonstrating when coarse particles materially affect plume microphysics (a simple calculation comparing potential water uptake per coarse particle to plume-available vapor is recommended).

9. Weighted-mean scaling is presented without conditions for linear applicability. The linear weighted-mean reconstruction around Eq. (3) is supported empirically but without a mathematical criterion (e.g., small-depletion limit, timescale separation, or inequality) that explains when nonlinear vapor-competition effects can be ignored.

10. The use of an isoline ($\max(J_{\mathrm{HDN}}) = 10^6\,\mathrm{m}^{-3}\mathrm{s}^{-1}$) in the HDN discussion remains qualitative because $J$ (which is a rate) is not integrated into expected droplet number or vapor removal over plume lifetimes.

11. When mentioning the planned neural-network parametrization (Part 3), briefly discuss how physical constraints (e.g., monotonicity, conservation) will be preserved.

12. To facilitate usage by third parties and accelerate the community's understanding of the problem, could the authors clarify the availability of the model used for these simulations (in the original papers, it is stated that the data are available upon request to the corresponding author)? Are there plans to release the model as open-source? I'm saying this because the paper rests on the running of a model. If the model would be open-source, then it is accessible to the entire scientific community, facilitating the reproduction and intercomparison of results. The authors mention about the availability of the data upon request to the author, which is fair, though current practices within Horizon Europe and national science programs are moving into the direction of publishing the data and making them findable, accesible, interoperable, etc. Please, consider these aspects. In the end, this enhances the impact of the research and supports the sharing of knowledge.

---

## Author Comment (AC1)

Dear editor and reviewers,

We thank the editor for handling our manuscript, and we are grateful to both reviewers for their careful reading of the manuscript and for the constructive and thoughtful comments. We have thoroughly addressed all comments and revised the manuscript accordingly.

In the course of the reviews, we undertook a reorganization of the manuscript to improve clarity and structure. Following Reviewer 2's comments, we moved parts of the original Introduction into a new dedicated Background section in order to provide a clearer and less fragmented motivation to the topic. Based on Reviewer 1's comments, we consolidated all results into a single Results section instead of two separate sections. In addition, we have added an explicit Discussion section in which we assess the generality and limitations of our findings, as well as their relation to conventional kerosene contrails. We also included additional figures and supporting text to clarify key aspects of the analysis and to strengthen the physical interpretation of the results.

As a result of these changes, the manuscript now follows a classical structure of a scientific paper, which we believe facilitates navigation and comprehension for the reader. Details of the restructuring and all associated modifications are described explicitly in our point-by-point responses to the reviewers' comments. We also note that all responses refer to the revised manuscript using the new section, equation and figure numbering.

Importantly, the structural reorganization and the addition of explanatory text and figures do not affect the original scientific conclusions of the manuscript. The revisions serve solely to improve structure and the physical discussion of our results.

In this response document, reviewer comments are reproduced in black. Our replies are provided in green, and references to the manuscript are indicated by *'green italicized text in quotation marks'*.

**Reviewer 1 (RC1)**

**General comments:**

In this manuscript, the authors use box-model simulations to estimate ice crystal number concentrations at the end of the jet regime. These are performed for hydrogen combustion and therefore consider only the entrainment of ambient particles (no exhaust emissions). The authors derive conditions under which homogeneous droplet nucleation may compete with heterogeneous droplet formation (on (entrained) particles). They also conclude that the ambient particle size distribution can be treated using a weighted mean approach. Finally, they perform several sensitivity analyses.

Overall, I find this manuscript provides a clear overview of the key considerations for simulating contrail formation from hydrogen exhausts. It is particularly useful to see an assessment of homogeneous versus heterogeneous droplet formation. However, I find that the manuscript would benefit from being restructured, to highlight the main results and differentiate these from observations about the model. Other specific comments are listed below. Therefore, I would recommend publication after the below comments have been suitably addressed.

**Specific comments (in form: location-comment):**

Overall structure: the conclusions drawn in Sect. 5 of the manuscript are useful and well-defined. I find that there are two main results: (a) the derivation of a conservative boundary for the

importance of homogeneous nucleation of water droplets and (b) ambient aerosol properties and/or meteorological conditions that result in asymptotic ice crystal number concentrations.

In addition, the authors show that (c) omission of the coarse-mode population of ambient particles has a negligible impact on model outputs and (d) they can reduce model compute by applying a weighted mean as in Eq. (3). Although the latter results are undeniably useful for reducing the model complexity and associated simulation times, the manuscript does not clearly motivate whether these are as useful to the wider community as (a) and (b). Therefore, I would recommend that either (i) the authors move (c) and (d) to a methodology/model development section and present (a) and (b) as final results or (ii) provide more justification for the wider applicability and limitations of (c) and (d) and retain these in a results section.

Moreover, I suggest that after restructuring/modifying as above, the main results are presented under a "results" heading rather than "microphysical insights". Currently, the model development is difficult to disentangle from the results. Accordingly, adopting these changes would more effectively highlight the most applicable results of the study.

We have carefully considered both structural options suggested by the reviewer and conclude that option (ii) best reflects the scientific nature of the findings. While it is correct that the omission of coarse-mode particles and the weighted-mean approach can ultimately be used to reduce model complexity, we regard both findings as *physical results* rather than mere methodological simplifications. Their validity is grounded in underlying microphysical processes:

- Coarse-mode particles are far too few in number to exert a significant influence on the water vapor budget.

- The weighted-mean approach succeeds because it is the total number of entrained and activated aerosols that determines the nonlinear competition for available water vapor.

These points represent fundamental insights into contrail formation from hydrogen combustion, where multiple ambient aerosol populations may contribute to ice-crystal formation. They therefore merit presentation as scientific results rather than as a technical model speed-up.

We acknowledge that the first version of the manuscript may have placed too much emphasis on technical aspects and did not sufficiently highlight the physical mechanisms. To address this, we have added clarifying figures and strengthened the physical interpretation throughout the text. In particular, we now include panel (c) in Figure 2, which explicitly shows the negligible vapor uptake by coarse-mode particles.

To further elucidate the weighted-mean concept, we have added a new figure (Fig. 3) with accompanying explanatory text that illustrates how the weighted mean is constructed and why it performs well for hydrogen combustion scenarios.

Given these revisions, we agree with the reviewer that a unified *Results* section is more appropriate. We therefore reorganized the structure such that all main findings now appear in a single results section. Within this,

- Sect. 3.1 presents the findings related to multiple aerosol populations, including the coarse-mode and weighted-mean results,

- Sect. 3.2 discusses the importance of homogeneous droplet nucleation, and

- Sect. 3.3 summarizes the systematic investigation.

This restructuring separates model description from physical insights, while maintaining the scientific coherence of the manuscript.

Homogeneous droplet nucleation: using information in Figure. 6 and Table. 3, it appears that some simulations may be performed under conditions where homogeneous droplet nucleation competes with heterogeneous droplet formation on entrained ambient particles (i.e., $p_a$ = 400 hPa and $T_a$ = 210 K). However, it is not clear whether homogeneous droplet nucleation is included in the simulations in Sect. 4. Therefore, I would suggest that the authors provide more information on this.

For the presented results, the HDN process has not been implemented in the box model. We merely evaluate the nucleation rates in an offline approach in Sect. 3.2. To do so, we take time series of T and $RH_w$ from box model simulations where activation of ambient aerosol is the only ice formation pathway. Based on this conservative offline analysis, we assess under which conditions HDN can safely be neglected.

HDN is a highly non-linear process, and its numerical implementation is challenging (see, e.g., Starzmann et al. (2018) for simulations of HDN in a different context). The integration of HDN into our box model will be described in another publication focusing on contrails from hydrogen fuel cell aircraft.

Based on the offline analysis, we only draw conclusions about the region of parameter space in which HDN can safely be neglected. Whether HDN actually becomes important in some of the scenarios explored in Sect. 3.3 (former Sect. 4), where it may potentially play a role, cannot be assessed within the present framework and requires simulations in which HDN is included online. To avoid ambiguity, we have now clarified this explicitly in the revised manuscript by adding
*'In addition, a small subset of temperature–pressure combinations in the dataset falls into a regime where the estimate presented in Sect. 3.2 suggests that HDN cannot be excluded as a potential contributor to ice formation. As HDN is not included in the simulations, the corresponding ice crystal numbers for these conditions are therefore subject to higher uncertainty.'*

Data availability: I would recommend that the authors consider putting figure data and/or underlying code in a public repository (e.g., Zenodo) rather than only being obtainable from the corresponding author.

We have created a Zenodo repository in which all data used in this study, as well as the plotting scripts required to reproduce the figures, are openly available. The repository is linked in the Data Availability section of the revised manuscript.

**Various:**

Line 33: this statement would benefit from additional references in support of hydrogen use in aviation. Currently, I find that the reference from Airbus is insufficient to motivate the potential for hydrogen uptake in this sector.

We agree with the reviewer that relying solely on the Airbus reference is insufficient to motivate the potential use of hydrogen in aviation. Industrial timelines and announcements can shift and therefore do not provide a stable scientific basis. For this reason, we removed the Airbus reference and instead added a broader and more enduring overview of conceptual propulsion configurations proposed for hydrogen use in aviation. We added
*'Two concepts are being explored for a potential future use of hydrogen in aviation (Tiwari et al., 2024; Soleymani et al., 2024; Richardson, 2025). One is hydrogen fuel-cell propulsion, where the fuel's chemical energy is converted into electricity that drives propeller-based systems. The other is*

*direct hydrogen combustion in gas turbines, which provide thrust either through a propeller (turboprop) or a high-velocity exhaust jet (turbofan). Propeller systems operate most efficiently at lower speeds and altitudes and could therefore be suited to regional or short-haul flights, whereas turbofan engines enable higher cruise speeds and altitudes and might also be applied to medium- and long-haul missions.'*

Lines 97 and 100: while the simulations performed by Yu et al., (2024) show the importance of fuel sulfur, the mass of volatile particles is a sum over the masses of all condensable gaseous species, which also include other organic material (Kärcher et al., 2000).

Thanks for this hint. We have revised the text to clarify this as follows:
*' In addition, volatile particles originating from condensable gaseous species (sulfuric acid, nitric acid, organics) (Kärcher et al., 2000) may contribute to ice crystal formation at high supersaturations, i.e., at low ambient temperatures. A recent study by Yu et al. (2024) hints at the importance of the sulfur content in the fuel for volatile particle formation.'*

Line 254: this is an interesting observation and would benefit from a fuller explanation. Specifically, why is this the case provided aerosol is entrained and not emitted?

With the addition of the new Figure 3 and the accompanying descriptive text, we now better clarify why the weighted-mean approach works for entrained ambient aerosols in the hydrogen combustion case. In the discussion section (Sect. 4), we further explain why such an approach is not expected to be generally applicable to emitted particles in the kerosene combustion case.

Lines 255-262: It is unclear what the total particle size distribution is being used in these simulations. Are the three modes described using a common total hygroscopicity parameter? If so, presumably it would be possible to represent this trimodal distribution as the linear combination of three monomodal particle size distributions.

The total particle size distribution used in the simulations is indeed constructed as a linear combination of monomodal particle size distributions. As defined earlier in the manuscript, we define an aerosol population as a monomodal size distribution with a specified hygroscopicity. In our setup, each of the three geometric mean radii is combined with two different hygroscopicities (representing weakly and highly soluble particles), resulting in six aerosol populations. We have revised the text to make this clearer as follows:
*'To do so, we perform simulations incorporating in total six aerosol populations, obtained by combining three monomodal size distributions (geometric mean radii $r_{d,aer}$ = {2, 20, 200} nm) with two hygroscopicities ($\kappa_{aer}$ = {0.05, 0.5}), such that each population represents a monomodal size distribution with an assigned hygroscopicity'*

Line 359: as some readers will not be familiar with the Koop parameterization, I would suggest that you explain the link between water-supersaturated conditions and the formation of liquid clouds more explicitly.

We reformulated the sentence and removed the reference to Koop, as it was misleading because the Koop parameterization describes freezing rather than droplet activation. The new sentence is:

*,If the environment were supersaturated with respect to water, natural clouds would be anyway present. '*

Line 381: although the temperature dependence is limited for soot-rich exhausts, more recent work (incorporating volatile particles) has shown that ambient temperature regulates contrail ice crystal number concentrations in the soot-poor regime (Yu et al., 2024).

We agree with the reviewer and have clarified this point by explicitly referring to the '*soot-rich regime*' in the revised manuscript.

Line 405: this discussion would benefit from additional (supplementary) information on the choice of freezing parameterization adopted in this work.

The freezing parameterization used in this study is described in detail in Appendix B of Bier et al. (2024). To improve clarity, we have now added a short summary of this parameterization to the model description (Sect. 2.3), including relevant references.
We included the following section:
'*The employed parameterization for homogeneous freezing (described in Appendix B of Bier et al. (2024)) accounts for the dependence of the freezing process on droplet water volume and cooling rate (Kärcher et al., 2015), as well as for freezing-point depression caused by dissolved solutes (Koop et al., 2000; O and Wood, 2016). The employed parameterization shows a strong decrease in the freezing fraction toward 235 K, consistent with laboratory experiments (e.g., Murray et al., 2010; O and Wood, 2016; Tarn et al., 2021; Ponsonby et al., 2024). We therefore apply a temperature threshold at 235 K and assume that supercooled droplets do not freeze into ice crystals for $T_a \geq 235K$.*'

Figure 8: I would advise clarifying (in both the figure and the main text) that the different colours are being used to represent different co-parameters and that respective ranges are taken from Table. 3.

We added '*Different colors indicate the various co-parameters, with the corresponding parameter ranges taken from Table 2.*' in the figure caption and '*…where the color denotes the co-parameter used for grouping*' in the main text.

**Minor comments:**

Line 13: using the results in Figure. 9, I would advise providing some quantitative information in the abstract to bound this regime (e.g., the information in line 409).

This is a valid point. We considered adding quantitative bounds to the abstract. However, doing so would exceed the abstract length constraint (<250 words). We therefore provide quantitative bounds for this regime in the Conclusion, such that the Abstract and Conclusion together give a coherent and sufficiently constrained description of the results.

Line 31: I would advise using the term "flight distance" rather than "range".

Thanks, we changed it accordingly.

Line 73: I would suggest revising the phrasing: "ready to be implemented".

Changed to "*and has been implemented in ECHAM-CCMod in the meantime*"

Lines 77-78 (and Figure 1 caption): I would suggest revising the order of "plume partial water vapor pressure" to "partial pressure of water vapor in the plume".

We changed it accordingly.

Line 80: I would suggest expanding on the significance of "lower calorific value" in this context or removing altogether.

Depending on the scientific community, the term combustion heat might be interpreted differently. Hence, we clarify that we speak about the lower calorific value and not the higher calorific value (see https://en.wikipedia.org/wiki/Heat_of_combustion).

Line 88: I would suggest clarifying that this is only true if the ambient conditions are the same in both cases.

We added *'…for the same ambient conditions'*.

Lines 91-92: I think it is more important to mention that ambient upper-tropospheric temperatures are usually below the Schmidt-Appleman temperature for hydrogen combustion. The comparison with the homogeneous freezing threshold is of secondary importance.

We respectfully disagree. If the Schmidt–Appleman criterion is usually fulfilled for hydrogen combustion under upper-tropospheric conditions, it no longer represents the limiting factor for contrail formation. Then, the decisive criterion becomes whether the activated droplets subsequently freeze into ice crystals, making the comparison with the homogeneous freezing threshold physically more relevant.
Climatological analyses of whether the thermodynamic Schmidt-Appleman critical temperature or the freezing temperature are the limiting factor in contrail formation have been addressed in several recent papers (Kaufmann et al., 2024; Hofer et al., 2024; Megill and Grewe, 2025).

Lines 93-94: I would suggest expanding on why the maximum plume supersaturation is important for the process of contrail formation.

We rewrote this sentence to '*Nevertheless, the extent to which the ambient temperature is below the Schmidt-Appleman threshold temperature still plays an important role, as it determines both the maximum plume supersaturation and the duration of supersaturated conditions, thereby controlling the activation of aerosol particles into liquid droplets.*'

Line 143: I would suggest expanding on the meaning of the term "simple enough". For instance, does this refer to total computation time?

We have clarified the meaning of "simple enough" by rewriting the sentence to '*…that is simple enough to be easily applied in other models (level of complexity, computing time, ease of implementation).*'

Line 148: I would suggest revising this sentence for clarity.

We appreciate the suggestion to improve clarity. However, as the comment does not specify which part of the sentence is unclear, we were unable to identify a concrete revision that would improve it and therefore left the sentence unchanged.

Line 175: I would advise defining the meaning of "phase" in this context.
We have clarified the meaning by adding '*…particle phase (dry, liquid, or ice)*'

Line 221: I would suggest replacing "spectrum" with "particle size distribution".
Thanks. We changed it accordingly.

Figure 2: please could you expand (in the text) on why $N_{ice,f}$ shows a slight decrease for large x-values rather than asymptotic behaviour.

This is caused by the freezing-point depression in solution droplets with large dry radii, an effect that is more pronounced for highly soluble particles (kappa = 0.5, left panel) than for weakly soluble ones (kappa = 0.05, right panel). This is explained in Sect. 3.3.2 (see also Fig. 9e). The purpose of Figure 2 is to illustrate the role of coarse-mode particles and their negligible indirect influence on the vapor budget; adding a detailed discussion of freezing-point depression at this point would distract from the main message of the figure.

Line 274: it would be useful to clarify the units of $J_{aer}(t)$ in Equation (4).

We agree that clarification is helpful. We have now explicitly stated the units $m^{-3}\,s^{-1}$ directly after the equation.

Line 374: I would suggest introducing a reference for the cutoff at 235 K. Alternatively, this could be enveloped in a supplemental section that outlines your homogeneous ice nucleation parameterization (as described earlier).

We agree that a reasoning for the 235 K cutoff is important. However, we consider this information to be more appropriately placed in the model description rather than late in the manuscript. We have therefore moved the statement to Sect. 2.3, where we now describe the homogeneous freezing parameterization and provide the relevant references. We added:
*'The employed parameterization shows a strong decrease in the freezing fraction toward 235 K, consistent with laboratory experiments (e.g., Murray et al., 2010; O and Wood, 2016; Tarn et al., 2021; Ponsonby et al., 2024). We therefore apply a temperature threshold at 235 K and assume that supercooled droplets do not freeze into ice crystals for $T_a \geq 235$ K.'*

Lines 350-354: I would suggest explaining this caveat in more detail as the implication (of the final sentence of this paragraph) is unclear to me.

We clarified it by adding '*Thus, even the less likely combinations represent extrapolations that still capture the primary influence of ambient pressure on the contrail formation process and do not deteriorate the representation of contrail formation under more realistic atmospheric conditions.*'

**Technical suggestions (in form: location-comment):**

Line 2: "well studied" should be hyphenated.

Hyphenation is not required in this case because "well studied" is used in predicative position (after the verb), and compound modifiers with "well" are only hyphenated when placed before a noun.
In any case, we revised the abstract to meet the word-limit constraint, and this phrasing no longer appears.

Line 60: I would suggest replacing "give meteorological condition" with "given *set of* meteorological condition*s*".

Many thanks. We changed it accordingly.

Line 63: "globally distributed" should be hyphenated

Done.

Line 76: there is a missing "the" in "(called *the* plume)".

Thanks.

Line 77: "Micro-physical" should not be hyphenated.

Changed.

Lines 77-78 (and Figure 1 caption): I would suggest revising the definition "plume partial water vapor pressure" as "partial pressure of water vapor in the plume".

Done (this comment was already raised under minor comments).

Line 121 (and line 210): I would suggest replacing "well" with "highly" and removing the hyphen.

Done.

Line 155: I would suggest revising the word order, particularly the use of "used".

We rephrased it to *'…a short review of the contrail formation model employed in the study'*

Line 176: "analytically prescribed" should be hyphenated.
Really? There is no noun that is modified.

Line 178: there is a typographic error in this sentence between "calculated" and "time-resolved".

Changed.

Line 205: I would suggest removing "already".
Thanks done.

Line 265: I would suggest rephrasing the "curse of dimensionality".

The term '*curse of dimensionality'* is typically used to describe a phenomenon like ours, see the Wikipedia page for an extended explanation
(https://en.wikipedia.org/wiki/Curse_of_dimensionality)
We also added a reference to a book of Richard Bellman who coined the term in the 1960ies.

**References:**

Kärcher, B., Turco, R. P., Yu, F., Danilin, M. Y., Weisenstein, D. K., Miake-Lye, R. C., and Busen, R.: A unified model for ultrafine aircraft particle emissions, J. Geophys. Res., 105, 29379–29386, https://doi.org/10.1029/2000JD900531, 2000.

Yu, F., Kärcher, B., and Anderson, B. E.: Revisiting Contrail Ice Formation: Impact of Primary Soot Particle Sizes and Contribution of Volatile Particles, Environ. Sci. Technol., 58, 17650–17660, https://doi.org/10.1021/acs.est.4c04340, 2024

Megill, L. and Grewe, V.: Investigating the limiting aircraft-design-dependent and environmental factors of persistent contrail formation, Atmos. Chem. Phys., 25, 4131–4149, https://doi.org/10.5194/acp-25-4131-2025, 2025.

Hofer, S., Gierens, K., and Rohs, S.: Contrail formation and persistence conditions for alternative fuels, Meteorol. Z., 33, 43–49, https://doi.org/10.1127/metz/2024/1178, 2024

Kaufmann, S., Dischl, R., and Voigt, C.: Regional and seasonal impact of hydrogen propulsion systems on potential contrail cirrus cover, Atmos. Environ., 24, 100 298, https://doi.org/10.1016/j.aeaoa.2024.100298, 2024

Starzmann, J., Hughes, F. R., Schuster, S., White, A. J., Halama, J., Hric, V., Kolovratník, M., Lee, H., Sova, L., Stastny, M., Grübel, M., Schatz, M., Vogt, D. M., Patel, Y., Patel, G., Turunen-Saaresti, T., Gribin, V., Tishchenko, V., Gavrilov, I., Kim, C., Baek, J., Wu, X., Yang, J., Dykas, S., Wroblewski, W., Yamamoto, S., Feng, Z., and Li, L.: Results of the International Wet Steam Modeling Project, Proceedings of the Institution of Mechanical Engineers, Part A: Journal of Power and Energy, 232, 550–570, https://doi.org/10.1177/0957650918758779, 2018.

The authors perform a comprehensive suite of Lagrangian Cloud Model (LCM) simulations to study ice nucleation on ambient aerosols in hydrogen combustion plumes. They quantify the negligible influence of coarse-mode aerosols, propose a weighted-mean scaling to approximate multi-population behavior, and derive a conservative boundary (Eq. 8) to determine when HDN can be neglected. Furthermore, they identify a parameter subspace (temperature ≤ 225 K, 10–100 nm aerosol size) where ice number formation ($N_{ice,f}$) becomes insensitive to aerosol properties, thus facilitating parametrization.

**General comments**

The manuscript is scientifically valuable, presenting a thorough microphysical modelling study of contrail formation for hydrogen combustion and explores important conceptual simplifications for parametrization development. However, issues regarding the structure, novelty, model validation, assumptions, and clarity of applicability need to be addressed before publication. Specifically, the work is well executed, carefully documented, and useful for the contrail/aviation community.

 However, it mostly reports comprehensive application of an existing particle-based Lagrangian Cloud Module and a large parameter sweep; it does not introduce a clear, novel physical mechanism in atmospheric chemistry or cloud microphysics. As such, in its present form it reads as an important and careful modeling/data paper but falls short of the level of novel atmospheric physics/chemistry unless the authors: (a) clarify and amplify the particular microphysical insight(s) that are genuinely new; (b) add further validation/uncertainty quantification; and (c) improve the argument that the identified insensitive subspace is a new physical result rather than an empirical property of this model/setup. As a consequence, the authors should note that splitting the research into 3 manuscripts is acceptable only if each part contains a clear, stand-alone novelty claim. As written, Part 1 is a thorough model-based analysis, and may be regarded as insufficiently novel for ACP on its own. We elaborate this general comment with further specific comments below:

We thank the reviewer for the thoughtful and detailed comments. We appreciate the depth of microphysical expertise reflected in the review and welcome the opportunity to integrate these insights into the manuscript. We view the scientific depth of the comments as an indication of the scientific relevance of the manuscript rather than as evidence of it being a simple or trivial model application without novelty character. We address all points raised in detail in the responses below.

**Specific comments**

1. Scope and Contribution: While the manuscript is scientifically valuable and the discussion of results is thorough, the novelty appears circumscribed to systematically running an existing model (Bier et al., 2024). It feels more like a follow-up study rather than a substantial, standalone modelling contribution. The authors mention splitting the analysis into three parts. I am unconvinced that the current contribution is sufficient to justify a trilogy of papers. I strongly suggest the authors consider compressing the work into one or two meaningful papers. For instance, the third part (building an AI-based surrogate model) seems to offer the distinct modelling contribution that is currently lacking in this first part; integrating these findings could significantly strengthen the publication.

We respectfully disagree with the assessment that the novelty of the manuscript is limited to running an existing model without new insights. Several central components of our analysis

represent new scientific contributions that were neither included nor anticipated in Bier et al. (2024):

- The role of coarse-mode particles was not examined in Bier et al. (2024) and is assessed for the first time in this study.
- Homogeneous droplet nucleation was not discussed nor mentioned in Bier et al. (2024). Therefore, its analysis here provides a substantially new contribution.
- The concept of the weighted mean is entirely new. In the revision, we more clearly articulate the physical rationale that underpins this approach.
- The general statements we derive are only possible due to the systematic exploration conducted in this study. The conclusion regarding the subspace in which the final number of ice crystals formed becomes nearly independent of ambient relative humidity, as well as of the size and solubility of ambient aerosols, was not part of the conclusions in Bier et al. (2024). This statement is based on physical reasoning and is not a side effect of the model setup.

We therefore view the present manuscript as a standalone contribution that advances microphysical understanding, rather than merely applying an existing model without new insights. In our view, novelty in atmospheric modelling research does not arise solely from adding new code components; it can equally emerge from new physical interpretations, new insights, and new conceptual frameworks. This manuscript provides such contributions.

We acknowledge that the first version of the manuscript was, in places, written in a rather technical manner and did not elaborate sufficiently on the underlying physical processes. In the revised manuscript, we now place greater emphasis on the physical interpretation and the broader scientific implications, extending the text where appropriate and adding supportive figures (see comments below).

Finally, we note that no other reviewer of Part 1 or Part 2 raised concerns regarding insufficient novelty or the overall structure of the planned trilogy. Nonetheless, we have carefully considered this feedback and have further strengthened the articulation of the manuscript's scientific contribution.

2. Introduction and Motivation: The introduction is currently too long and fragmented. The critique of the state-of-the-art is not sufficiently connected to the specific scientific contributions of this study. Furthermore, the motivation needs to be more sharply focused on the aviation industry's context.

We agree with the reviewer that the introduction in the original manuscript was somewhat fragmented. In the revised version, we have therefore reorganized the structure: the review of contrail formation for hydrogen combustion and the overview of ambient aerosol properties have been moved to a dedicated section 'Background and model description' (new Sect. 2). The introduction now begins with a clearer and more concise motivation. The discussion of the contrail lifecycle has been shortened to place greater emphasis on the formation stage. In addition, we have expanded the passage on the potential use of hydrogen in aviation, thereby sharpening the contextual motivation of the study.

3. Outdated Context: The reliance on the Airbus (2020) reference for motivation feels outdated. In 2020, the target for service entry was around 2035. However, as of 2025, timelines have shifted

towards 2040–2045 due to infrastructure challenges (e.g., lack of hydrogen infrastructure at airports). Indeed, I have the feeling that the H2 aircraft is not a priority anymore... In any case, please, try to reflect the current industrial reality to ensure the motivation is robust and up-to-date.

We appreciate the reviewer highlighting the importance of an up-to-date and robust motivation. We agree that industrial timelines and priorities can shift, and relying on specific industry announcements may therefore not provide a stable scientific basis. Scientific investigations, however, aim to address fundamental questions that remain relevant beyond changes in industry focus.  Indeed, studies of contrails from hydrogen aircraft have been published for more than two decades (e.g., Ström and Gierens, 2002, Marquart et al., 2005, Ponater et al., 2006), demonstrating that the topic is not tied to any particular industrial timeline.

In light of this, we have removed the reference to the Airbus (2020) announcement from the introduction. Instead, we now provide a short and more general description of conceptual propulsion configurations investigated for the potential use of hydrogen in aviation. We added *'Two concepts are being explored for a potential future use of hydrogen in aviation (Tiwari et al., 2024; Soleymani et al., 2024; Richardson, 2025). One is hydrogen fuel-cell propulsion, where the fuel's chemical energy is converted into electricity that drives propeller-based systems. The other is direct hydrogen combustion in gas turbines, which provide thrust either through a propeller (turboprop) or a high-velocity exhaust jet (turbofan). Propeller systems operate most efficiently at lower speeds and altitudes and could therefore be suited to regional or short-haul flights, whereas turbofan engines enable higher cruise speeds and altitudes and might also be applied to medium- and long-haul missions.'*
This broader framing avoids reliance on specific industry statements and strengthens the long-term scientific motivation of the study.

 4. The study is purely model-based, with no quantitative validation. Although hydrogen contrails are not yet observed, consistency checks with kerosene contrails or known ranges of $N_{ice,f}$ would be recommended. Alternatively, explicitly state the limits of observational validation for hydrogen combustion and discuss implications for model uncertainty.

We appreciate the reviewer's emphasis on the challenges of quantitative validation for hydrogen contrails. The reviewer is correct that observational data for hydrogen combustion are currently unavailable, which inherently limits direct validation of hydrogen-specific microphysical processes.

Bier et al. (2022) compared the LCM box model to other contrail formation models for kerosene combustion, although such comparisons cannot validate the microphysical processes specific to hydrogen combustion. In addition, Bier et al. (2024) reported a comparison of $N_{ice,f}$ for hydrogen and kerosene combustion. We now cite and summarize these findings in Sect. 2.1 by adding *'Bier et al. (2024) showed that, at the same ambient temperature and pressure and for typical aerosol number concentrations (Sect. 2.2), the resulting ice crystal number is reduced by roughly one to two orders of magnitude compared to soot-rich kerosene combustion cases.'*

To address the reviewer's suggestion, we have added a dedicated discussion section (new Sect. 4) in which we outline the limitation of observational validation, discuss the implications for model uncertainty, and relate the hydrogen results to established knowledge from kerosene contrails.

5. Include sensitivity runs or describe variability when other subsets or entrainment efficiencies are used (now, the entrainment efficiency seems to be fixed in the paper).

We thank the reviewer for raising the question of the dependence on dilution speed (the entrainment efficiency/rate is simply the negative logarithmic derivative of the dilution factor). The reviewer is correct that we did not place strong emphasis on this aspect in the present Part 1. Many earlier and frequently cited contrail-formation studies have entirely neglected the influence of dilution speed (e.g., Kärcher and Yu, 2009; Yu et al., 2024). For this reason, we chose to give this aspect dedicated focus. Part 2 provides an extensive analysis of dilution effects, including a physically motivated discussion of how engine parameters influence the entrainment rate. Part 2 also introduces a scaling relationship that quantifies how dilution speed affects the final ice crystal number (slower dilution leads to lower $N_{ice,f}$ because earlier-entrained aerosols have more time to deplete the water vapor and thereby suppress activation of later-entrained aerosols). Explaining these mechanisms in detail requires substantial space, which also justifies the multi-part structure.

Importantly, although dilution speed modifies the total number $N_{ice,f}$, the main conclusions drawn in the present Part 1 remain valid. The weighted-mean approach still applies because it is the total number of entrained aerosols that governs the nonlinear competition for water vapor (see response to comment 9). The conclusions regarding the indirect effects of coarse-mode particles are also unaffected: their number concentrations are typically far too low to significantly alter the water-vapor budget. The assessment of the importance of HDN is also robust, as it is based on a conservative offline estimate using order-of-magnitude considerations (we only state where HDN can be safely neglected (see response to comments 6,7, 10)).

To avoid unnecessarily expanding the scope of Part 1 or distracting from its central conclusions, we do not include sensitivity studies in the main manuscript. Nevertheless, we now discuss this aspect explicitly in the discussion section (Sect. 4). For completeness in this review response, we provide supporting figures:

- **Figure 1** shows simulation results in which dilution is accelerated (time is scaled by the factor $s_{dil}$=0.5) or slowed down ($s_{dil}$ =2). While the total ice-crystal number changes, the weighted-mean approximation remains accurate. We note that a factor-of-two change in dilution speed would typically be associated with substantial changes in engine size or jet speed (see Part 2).

- **Figure 2** reproduces Figure 2 of the main manuscript but with slowed-down dilution ($s_{dil}$ =2), allowing early-entrained coarse-mode particles more time to consume water vapor. The conclusion remains unchanged: few coarse-mode particles do not exert an indirect effect on $N_{ice,f}$ !

[Figure]

*Figure 1: Evolution of (a) plume relative humidity and (b) number of ice crystals for different dilution speeds. In (b) the weighted mean approximation is also shown.*

[Figure]

*Figure 2: Reproduction of Figure 2 in the main manuscript but for slowed-down dilution.*

6. Criterion for HDN relevance (Eq. 8) should be further discussed. The conservative criterion $\max(J_{HDN, n_{aer} = 0} = 1\mathrm{e}6\ \mathrm{m}^{-3}\,\mathrm{s}^{-1})$ is useful but requires clearer justification. Specifically, it is

recommended to explain the physical rationale for this threshold and demonstrate how outcomes vary if it changes by an order of magnitude.

We first clarify that we do **not** interpret the criterion in Eq. (9) (former Eq. (8)) as a sharp threshold separating regimes in which HDN is irrelevant below the boundary and dominant above it. Rather, the offline estimate is explicitly designed to identify regions of parameter space in which HDN can **safely be neglected**.

This interpretation was already stated in the original manuscript, where we write:
*'If a $p_a$–$T_a$ pair lies to the right/below of this boundary (white region in Fig. 7b), we can be very sure that HDN does not play a significant role as the nucleation rates on the ambient aerosols are substantially higher regardless of the ambient aerosol number concentration. In the gray region of Fig. 7b, it remains uncertain whether HDN may significantly contribute to droplet formation. This depends on the aerosol number concentration and requires an explicit, time-resolved simulation of relative humidity and temperature evolution, along with the kinetics of droplet cluster growth/evaporation.'*

The estimate underlying Eq. (9) is intentionally conservative and described in detail in the manuscript. The chosen threshold is selected such that, even under extremely clean atmospheric conditions, the contribution of HDN remains negligible compared to nucleation on ambient aerosols (see also our response to Comment 10). In Fig. 7b, we marked the conditions corresponding to the exemplary time evolutions shown in Fig. 6. The conditions used for the right-hand column of Fig. 6 lie slightly to the right of the conservative boundary. The time evolution of nucleation rates (Fig. 6d) clearly demonstrates that HDN can safely be neglected under these conditions.

Many scenarios within the gray region of Fig. 7b are still expected to exhibit negligible HDN contributions, particularly at higher ambient aerosol number concentrations. This is because our conservative estimate intentionally overestimates HDN rates by evaluating them conservatively along the mixing line, thereby neglecting the water-vapor depletion caused by already activated aerosol particles (see Fig. 6c). Figure 7b also shows alternative isolines corresponding to different threshold choices; we deliberately adopt a very conservative isoline, for which we can be confident that HDN can be neglected below it. Drawing quantitative conclusions within the gray region would be speculative without explicit, time-resolved simulations including HDN.

Overall, the offline estimate demonstrates that HDN may only become relevant at very low temperatures, while in many scenarios it can be safely neglected.

7. Further discuss/quantify how neglecting vapor depletion or droplet interactions may overestimate HDN frequency. The exclusion of lubrication-oil or NOx-product aerosols may limit generality. In this respect, it is recommended to discuss how their presence could modify the current conclusions, especially the negligible coarse-mode effect.

As described in our response to Comment 6, our analysis is explicitly designed to draw conclusions only for the white region in Fig. 7b, where HDN can safely be neglected. The reviewer's question primarily concerns the gray region, for which we deliberately refrain from making quantitative statements in order to avoid speculative conclusions.

HDN rates are overestimated in our conservative estimate because water-vapor depletion by activated ambient aerosol particles (Fig. 6c,d) and interactions among HDN-formed droplets are neglected. Properly capturing these effects requires fully time-resolved simulations with HDN included online; attempting to quantify them using offline estimates would therefore be of limited

value and potentially misleading. The numerical implementation of HDN in our box model will be presented in a separate publication focusing on contrails from hydrogen fuel cell aircraft.

The presence of additional particle types (e.g., lubrication-oil or NOx-product aerosols) would further deplete water vapor and thus make HDN even less likely. Consequently, the conservatively estimated white region remains valid and general. The same argument applies to coarse-mode particles: since they already have a negligible influence on the water-vapor budget when only ambient aerosols are considered, their impact remains negligible if additional particle types contribute to vapor depletion.

All these aspects are now also discussed in Sect. 4 of the manuscript.

8. The paper neglects the coarse-mode particles without quantified depletion analysis. The manuscript asserts these particles can be ignored when they are orders of magnitude less numerous, but provides no order of-magnitude vapor budget demonstrating when coarse particles materially affect plume microphysics (a simple calculation comparing potential water uptake per coarse particle to plume-available vapor is recommended).

As stated in the manuscript, coarse-mode particles are typically several orders of magnitude lower in number than co-existing other populations. The aim of our analysis was therefore not to assess their direct impact on $N_{ice,f}$, but rather to determine whether they exert any indirect influence through significant depletion of available water vapor.

Our results show that this indirect effect is negligible. To clarify this, we have now included panel (c) in Figure 2 of the main manuscript, which explicitly displays the water-vapor uptake by coarse-mode particles. The plot demonstrates that their contribution to vapor depletion is negligible.  We wrote *'Due to their low relative abundance and the continuous entrainment of aerosols into the plume over time, they are not abundant enough to significantly deplete the water vapor at an early stage to hinder the smaller-sized particles in their activation. Indeed, the amount of water vapor taken up by the few coarse mode particles is negligible compared to the total water vapor available in the plume (emitted water-vapor mass plus entrained ambient moisture) during droplet and ice crystal formation (Fig. 2c).'*

This conclusion is not an empirical property of a specific model configuration; it follows directly from the diffusional growth equations for condensation and deposition. The available growth time is simply too short for coarse-mode particles to grow large enough to affect the vapor budget. As shown in the supporting figure provided above (see response to comment 5), this remains true even under slower dilution conditions that provide more time for droplet and ice crystal growth.

In summary, coarse-mode particles with low number do not require special treatment. In the light of the weighted-mean approach, coarse mode particles would become important if they were present in sufficiently large numbers for their weight to become non-negligible, i.e., when they exert a direct effect. Under typical atmospheric conditions, this is not the case. Therefore, we neglected them in the subsequent analyses.

 9. Weighted-mean scaling is presented without conditions for linear applicability. The linear weighted-mean reconstruction around Eq. (3) is supported empirically but without a mathematical criterion (e.g., smalldepletion limit, timescale separation, or inequality) that explains when nonlinear vapor-competition effects can be ignored.

From this comment we deduce that the physical basis for the success of the weighted-mean approach was not explained sufficiently in the first draft. The key point is that the weighted mean is constructed from simulations in which each aerosol population is included individually but always with the **total aerosol number concentration**. This design ensures that water-vapor competition—fundamentally controlled by the total number of already entrained and activated particles—is consistently represented in each component simulation.

To clarify this concept, we have added a new figure to the revised manuscript (Fig. 3), accompanied by descriptive text that explains how the weighted-mean approach is constructed and why it performs well in the hydrogen combustion case. Strictly speaking, the weighted-mean approach is not a linear operation, and its success cannot, for example, be attributed to a small-depletion limit. If such a limit were the governing mechanism, then simply summing simulations in which each population is included with its actual number concentration would be adequate. The new figure demonstrates that this procedure does **not** yield accurate results.

Figures 4 and 5 show that the weighted mean approach works across a wide variety of scenarios (different aerosol properties, ambient temperatures, total number concentrations, and up to six aerosol populations). These examples demonstrate that the success of the weighted mean is **not** an empirical finding tied to a particular model setup, but rather follows directly from the physical observation that the **total number of already entrained and activated aerosol particles drives the nonlinear competition for the available water vapor**. For this competition, other properties of the aerosol particles are of minor importance.

In the revised discussion section (Sect. 4), we also explain why we do not expect such an approach to be generally applicable to scenarios involving emitted particles in kerosene combustion.

10. The use of an isoline $\max(J_{HDN, \, naer=0} = 1e6 \ m^{-3} \ s^{-1})$ in the HDN discussion remains qualitative because J (which is a rate) is not integrated into expected droplet number or vapor removal over plume lifetimes.

We agree that J is a nucleation rate. Based on the exemplary time evolutions shown in Fig. 6, we considered the maximum nucleation rate to be a meaningful proxy for the integrated number of formed droplets. To make this relation more explicit and quantitative, we have now added an additional figure to the Appendix (Fig. B1) and added in the main text
*'…Although these values represent instantaneous rates, they serve as a robust indicator of the integrated total number (Appendix B).'*

Figure B1 shows that the maximum nucleation rate on entrained ambient aerosols scales nearly linearly with the ambient aerosol number concentration and that it indicates the order of magnitude of the integrated number. Moreover, under the assumption that plume temperature and relative humidity evolve along the mixing line, a nearly linear relationship between the maximum homogeneous nucleation rate and the integrated number is evident.

Importantly, Fig. B1 also demonstrates that for the conservatively chosen threshold $\max(J_{HDN, \, naer=0} = 1e6 \ m^{-3} \ s^{-1})$ the hypothetically integrated number of droplets formed by HDN—obtained under the assumption that plume temperature and relative humidity evolve along the mixing line—remains at least two orders of magnitude smaller than the integrated number of droplets formed on ambient aerosols, even for very low ambient aerosol number concentrations. This supports the use of the selected isoline as a conservative criterion for identifying conditions under which HDN can safely be neglected.

11. When mentioning the planned neural-network parametrization (Part 3), briefly discuss how physical constraints (e.g., monotonicity, conservation) will be preserved.

We thank the reviewer for raising this important point. The question of how physical constraints are preserved in the neural-network parametrization is indeed central. However, these aspects are explicitly addressed in Part 3, where the parametrization is actually presented. Describing the corresponding methods requires substantial explanation and therefore cannot be meaningfully integrated into the present Part 1 without shifting focus away from the microphysical analysis.

This consideration is one of the reasons motivating the three-part structure of the work. The study was intentionally divided into (i) microphysical process analysis (Part 1), (ii) engine-related and dilution aspects (Part 2), and (iii) the development of the final parametrization (Part 3). Each part addresses a distinct scientific component, and combining all elements into a single paper would exceed a balanced and coherent presentation.

12. To facilitate usage by third parties and accelerate the community's understanding of the problem, could the authors clarify the availability of the model used for these simulations (in the original papers, it is stated that the data are available upon request to the corresponding author)? Are there plans to release the model as open-source? I'm saying this because the paper rests on the running of a model. If the model would be open-source, then it is accessible to the entire scientific community, facilitating the reproduction and intercomparison of results. The authors mention about the availability of the data upon request to the author, which is fair, though current practices within Horizon Europe and national science programs are moving into the direction of publishing the data and making them findable, accesible, interoperable, etc. Please, consider these aspects. In the end, this enhances the impact of the research and supports the sharing of knowledge.

We have created a Zenodo repository in which all data used in this study, as well as the plotting scripts required to reproduce the figures, are openly available. The repository is linked in the Data Availability section of the revised manuscript.

The box model code contains legacy code parts for which the copyright situation is not clear. Hence, we need to work on these issues before we can publish the full code with a proper license.

Sharing the post-processing tools and the simulation data is uncritical in this regard and we gladly publish them.

**References:**

Marquart, S., Ponater, M., Ström, L., and Gierens, K.: An upgraded estimate of the radiative forcing of cryoplane contrails, Meteorol. Z., 14, 573–582, https://doi.org/10.1127/0941-2948/2005/0057, 2005

Ponater, M., Pechtl, S., Sausen, R., Schumann, U., and Hüttig, G.: Potential of the cryoplane technology to reduce aircraft climate impact: A state-of-the-art assessment, Atmos. Environ., 40, 6928–6944, https://doi.org/10.1016/j.atmosenv.2006.06.036, 2006.

Ström, L. and Gierens, K.: First simulations of cryoplane contrails, J. Geophys. Res., 107, 6, https://doi.org/10.1029/2001JD000838, 2002

Kärcher, B. and Yu, F.: Role of aircraft soot emissions in contrail formation, Geophys. Res. Lett., 36, L01 804, https://doi.org/10.1029/2008GL036649, 2009

Yu, F., Kärcher, B., and Anderson, B. E.: Revisiting Contrail Ice Formation: Impact of Primary Soot Particle Sizes and Contribution of Volatile Particles, Environ. Sci. Technol., 58, 17 650–17 660, https://doi.org/10.1021/acs.est.4c04340, 2024.